# Identification of *Mycoplasma* Species in Cattle Associated with Bovine Respiratory Disease Mortality

**DOI:** 10.3390/microorganisms12112340

**Published:** 2024-11-16

**Authors:** Emanuele Carella, Erika Messana, Davide Mugetti, Elena Biasibetti, Marzia Pezzolato, Simone Peletto, Mattia Begovoeva, Francesca Rossi

**Affiliations:** 1Istituto Zooprofilattico Sperimentale del Piemonte, Liguria e Valle d’Aosta, 10154 Torino, Italy; emanuele.carella@izsplv.it (E.C.); erika.messana@izsplv.it (E.M.); elena.biasibetti@izsplv.it (E.B.); marzia.pezzolato@izsplv.it (M.P.); francesca.rossi@izsplv.it (F.R.); 2Dipartimento di Prevenzione, Azienda Sanitaria Locale del Verbano Cusio Ossola, Via Mazzini 117, 28887 Omegna, Italy

**Keywords:** *Mycoplasma hyopharyngis*, BRD, *Mycoplasma* spp., *Ureaplasma* spp., seasonality, co-infections

## Abstract

Approximately 30 distinct *Mycoplasma* species have been isolated from cattle, but only a few are pathogenic and can cause serious respiratory diseases. Consequently, this study aimed to identify *Mycoplasma* spp. infections in cattle with bovine respiratory disease (BRD), considering factors such as animal demographics, concurrent infections with other pathogens, post-mortem clinical findings and histological examinations, and seasonality. A total of 326 samples were collected from 322 cattle that had died from BRD in Northwestern Italy. A total of 54 animals (16.8%) tested positive for *Mycoplasma* spp., and *Mycoplasma bovis* (*n* = 22, 40.7%) and *Mycoplasma dispar* (*n* = 13, 24.1%) were the most frequently detected species among the examined cattle. Among positive cattle, those aged five months or younger were approximately five times more likely to be infected by *Mycoplasma dispar* than by *Mycoplasma bovis* compared to those older than five months (proportional incidence ratio: 5.1, 95% CI 1.2–21.2). The main bacterial pathogens identified in cattle exhibiting co-infection was *Pasteurella multocida*, whereas the main viral pathogens were BRSV and BoHV-1. Histopathological investigations predominantly revealed catarrhal bronchopneumonia or purulent catarrhal bronchopneumonia among the examined cattle. Finally, *Mycoplasma hyopharyngis*, a species isolated from the pharyngeal and nasal cavities of pigs so far, was detected for the first time in the pneumonic lung of a bovine infected with BRD. Further investigations are necessary to thoroughly characterize its host range and pathogenic potential.

## 1. Introduction

The class Mollicutes comprises approximately 200 species across 14 genera, with representatives found ubiquitously in plants, animals, and humans. The genera *Mycoplasma*, *Ureaplasma*, and *Acholeplasma* are recognized to include species that inhabit animals as commensals, saprophytes, or pathogens [1]. Among the five genera within Mollicutes, *Mycoplasma* and *Ureaplasma* are associated with diseases in ruminants. Infections due to different *Mycoplasma* species can have significant implications for animal welfare and the economic viability of farming and livestock production industries worldwide [2]. Approximately 30 distinct *Mycoplasma* species have been isolated from cattle, with only a minority considered pathogenic and capable of causing severe respiratory diseases in both young and adult animals [3]. Several *Mycoplasma* species have been identified colonizing the bovine respiratory mucous membranes. Some of these species are pathogenic, leading to bronchopneumonia, while others are ubiquitous and form part of the normal microbiota [4,5]. *Mycoplasma* species can also exert a pathogenic influence on the reproductive system, as they are associated with diseases of bulls’ genital tract and reproductive disorders in cows, such as mastitis, metritis, abnormal vaginal discharge, and abortion [3,4]. Furthermore, *Mycoplasma* spp. have been associated with infectious keratoconjunctivitis, suppurative otitis media, meningitis, decubital abscesses, and endocarditis. These bacteria can disseminate through the bloodstream, colonizing diverse organs within the cattle host [5,6,7,8,9].

*Mycoplasma bovis* is a bacterial member of the family Mycoplasmataceae. It is the most frequently encountered *Mycoplasma* species in cattle in North America and Europe and the most important and commonly isolated *Mycoplasma* species associated with respiratory disease in cattle worldwide [4,10]. Traditionally, *M. bovis* has been mostly associated with chronic bovine respiratory disease (BRD), characterized by pneumonic lesions that often do not respond to antibiotic therapy [8,11]. BRD is a significant health issue for cattle globally, as it is a multifactorial disease involving pathogens, a compromised host immune system, and environmental factors [12]. Typical clinical signs of BRD include pneumonia, fever, dyspnea, coughing, nasal or eye discharge, depression, and anorexia. However, the clinical picture of respiratory disease is not usually characteristic and often does not differ from clinical signs caused by infections with other bovine respiratory tract pathogens, especially in the presence of co-infections [8]. BRD is frequently present in the cattle population, causing disease only if specific conditions occur. The animals’ immune response is impaired due to stress such as inadequate feeding, transportation, and environmental temperatures [13,14]. High in-barn temperature, frequent animal movement, moldy feed, overcrowded barns, and concomitant disease are the main stress-factors that may affect the immune system. Additionally, a high average milk production may indicate that cows were at a higher risk of being in a negative energy balance if not fed adequately and therefore might have been more susceptible to infectious disease [15]. Therefore, BRD represents an enormous animal welfare issue and an economic burden to beef cattle production [14]. In Europe, *M. bovis* is believed to be responsible for 25.0% to 33.0% of calf pneumonia cases [2]. Contrastingly, in the USA it is estimated that BRD morbidity is 16.2% in feedlots, while the mortality accounts for 55.0% and 36.0% in cattle and calves, respectively [11].

The primary viral pathogens associated with BRD include bovine herpesvirus 1 (BoHV-1), bovine viral diarrhea virus (BVDV), bovine parainfluenza virus 3 (BPIV-3), bovine coronavirus (BCoV), and bovine respiratory syncytial virus (BRSV). Although viruses generally exhibit mild clinical signs, they can impair defense mechanisms and immune system function, thereby increasing susceptibility to secondary bacterial infections [12]. Bacterial agents frequently associated with BRD include *Histophilus somni*, *Mannheimia haemolytica*, and *Pasteurella multocida* [12]. In addition to *Ureaplasma diversum*, several *Mycoplasma* species, including *M. dispar*, *M. bovigenitalium*, *M. bovirhinis*, *M. alkalescens*, *M. arginini*, *M. canadense*, and *M. canis*, have been implicated in the development of BRD. Some of these species are often found as part of the microbial microbiota of the upper respiratory tract in healthy calves. In most reports, they have been isolated in mixed infections with other known pathogens [4]. Co-infection with other causative organisms can intensify the severity of respiratory disease in calves, as well as increase morbidity and mortality rates, leading to a severe form of calf pneumonia [16]. Nevertheless, *M. bovis* has been most linked to BRD, characterized by pneumonic lesions that often do not respond to antibiotic therapy [11]. *Mycoplasma* infection is typically introduced into *M. bovis*-free herds by clinically healthy calves or young cattle. These animals may harbor *M. bovis* and shed the microorganism through their nasal discharge for months or even years [4].

The understanding of environmental sources of *Mycoplasma* spp. and their role in disease transmission remains limited. A small number of environmental sources, including cooling ponds, dry lots, and recycled bedding, have been identified as possible fomites where *Mycoplasma* spp. have been isolated on dairy farms. Houseflies may also pose a potential risk to dairy farms in terms of disease transmission [17]. In addition, the possibility of airborne transmission of *M. bovis* is not well-defined, with only a few findings supporting this infection route. When calves were exposed to aerosolized *M. bovis*, it resulted in respiratory disease. Despite the absence of clinical signs, specific *M. bovis* lung lesions were observed in the infected calves, as confirmed by necropsy and histological examinations. Moreover, the detection of *M. bovis* from the upper trachea in most of the infected calves provided further evidence of this infection route [8].

The concurrent infection of cattle with *Mycoplasma* and other pathogens can exacerbate the severity of respiratory diseases, leading to increased morbidity and mortality rates. This often results in a severe manifestation of pneumonia [16]. Beyond the respiratory system, *Mycoplasma* spp. can also exhibit pathogenicity in the reproductive tract and disseminate systemically within the host. These complications are the primary factors responsible for significant losses in the cattle industry. Consequently, this study aimed to identify *Mycoplasma* spp. infections in cattle with bovine respiratory disease (BRD), considering a wide range of factors such as animal demographics, concurrent infections with other pathogens, post-mortem clinical findings, histological examinations, and seasonality. This research therefore provides detailed insights into the proportional incidence, distribution, and potential impact of *Mycoplasma* spp. in BRD cases and may contribute to developing effective prevention and control strategies for this economically significant disease.

## 2. Materials and Methods

### 2.1. Sample Collection

A total of 326 samples were collected during necropsy from 322 cattle that died from BRD across 46 farms in Northwestern Italy, between January 2020 and June 2023. The collected biological matrices included 321 lung samples, 4 tracheal samples, and 1 pulmonary exudate. In 4 cases, both lung and trachea were obtained from the same animal, while the pulmonary exudate was collected along with the lung in a single bovine. Lungs, tracheas, and pulmonary exudate were collected and immediately stored at −80 °C for molecular, microbiological, and histopathological investigations. The flowchart in Figure 1 provides a clear and concise overview of the experimental design. Population data (sex, age, and sampling location) were recorded for all *Mycoplasma*-positive cattle, as accompanying veterinary reports provided detailed demographic information for each animal (Appendix A).

### 2.2. Identification of Etiological Agents

#### 2.2.1. Microbiological Investigations

Lungs and tracheas collected during necropsy were subjected to microbiological assays in accordance with the Standard Operating Procedure (SOP) to appropriately treat the samples for respiratory pathological patterns. A portion of lung or trachea exhibiting macroscopic lesions was initially cauterized and then streaked onto Columbia Blood Agar. (CBA) (Liofilchem Ltd., Roseto degli Abruzzi, Italy) and Gassner Agar (GA) plates (Microbiol & C., Cagliari, Italy) and chocolate agar medium (Liofilchem Ltd., Roseto degli Abruzzi, Italy). The samples were then incubated both overnight at 37 °C under aerobic conditions in an incubator PID system type M150-TB (MPM instrument srl, Bernareggio, Italy) and for 24 h at 37 °C under microaerophilic conditions (5.0% CO_2_) in a CO_2_ incubator MCO-17AIC (Sanyo, Osaka, Japan). CBA plates were also used for satellitism isolation procedures, with *Staphylococcus aureus* serving as the source of the V factor in the primary culture on CBA. After 48–72 h of incubation under microaerophilic conditions (5.0% CO_2_), suspected factor V-dependent colonies grew along the hemolytic area around the staphylococci growth.

Chocolate agar and Schaedler agar were also used to cultivate fastidious and anaerobic bacteria, respectively. Chocolate agar was incubated at 37 °C under microaerophilic conditions (5.0% CO_2_) in a CO_2_ incubator MCO-17AIC (Sanyo, Osaka, Japan) for 24–48 h to cultivate fastidious bacteria such as *Histophilus* spp. and *Corynebacterium* spp. Schaedler agar was incubated inside an anaerobic gas generating pouch system with an indicator for 24–48 h at 37 °C to cultivate anaerobic bacteria such as *Clostridium perfringens*.

The suspicious colonies obtained, originating from both routine microbiological practices and satellitism isolation procedures, were identified to the genus level using matrix-assisted laser desorption ionization-time of flight mass spectrometry (MALDI-TOF) (Bruker Daltonik GmbH, Billerica, MA, USA), according to the manufacturer’s instructions [18].

The search for fungal species was performed by inoculating organ samples onto Sabouraud dextrose agar plates, which were then incubated in a thermostat at 37 °C for at most 10 days. The fungal colonies were then stained with lactophenol blue and observed under a light microscope at progressive magnifications of 10× and 20×. For further confirmation, these fungal colonies were also subjected to sequencing.

#### 2.2.2. DNA Extraction and PCR Conditions for *Mycoplasma* spp.

DNA was extracted using the ExtractMe genomic DNA kit (Blirt, Gdańsk, Poland) following the manufacturer’s instructions. The detection of *Mycoplasma* spp. was performed by end-point PCR using the Invitrogen™ Platinum™ Quantitative PCR SuperMix-UDG kit (Invitrogen, Waltham, MA, USA). The primers were manufactured by Thermo Fisher Scientific (Waltham, MA, USA) and were specifically designed in previous studies [19,20]. The PCR mixture consisted of 2.5 μL of 10× buffer, 5 μM of forward primers (5′-ACTCCTACGGGAGGCAGCAGTA-3′), 5 μM of reverse primers (5′-TGCACCATCTGTCACTCTGTTAACCTC-3′), 10 mM of dNTPs, 50 mM of MgCl_2_, 5 U/μL of Taq polymerase, 2.5 μL of template, and nuclease-free water up to the final volume of 25 μL. The PCR cycling conditions on the 2720 Thermal Cycler (Applied Biosystems, Waltham, MA, USA) consisted of an initial step at 94 °C for 5 min followed by 35 cycles at 94 °C for 1 min, 58 °C for 90 s, and 72 °C for 1 min. Positive control for *Mycoplasma* spp. (CRFK cell line tested positive for *Mycoplasma* spp.) as well a negative control (distilled water), were added into our molecular analysis. Amplification products were separated on a 1.8% agarose gel by electrophoresis for 45 min at 120 V. The gel was then visualized using the AmpliSize Molecular Ruler (Bio-Rad, Hercules, CA, USA). Gel electrophoresis was used as a preliminary step to detect *Mycoplasma* spp. before sequencing, which is a crucial step to confirm the species of the detected *Mycoplasma*.

#### 2.2.3. Identification of *Mycoplasma* Species and Phylogenetic Analysis

After agarose gel electrophoresis, amplicons corresponding to *Mycoplasma* spp. were excised and separated from the gel. The excised gel bands were then purified using the Extract ME DNA clean-up & gel-out kit (Blirt, Gdańsk, Poland). Cycle sequencing was performed with forward and reverse primers using the Brilliant Dye Terminator 3.1 Cycle Sequencing Kit (NimaGen, Nijmegen, The Netherlands). Following purification, the samples were analyzed using the SeqStudio Genetic Analyzer (Applied Biosystems). Species identification was carried out by comparing the obtained sequences with those available on GenBank by Blast analysis using a similarity threshold of 97.0%. As a further confirmatory test, a phylogenetic tree was generated using MEGAX software (version 10.2.3), including *Mycoplasma* samples from this study and various reference sequences from GenBank. A bootstrap test of 1000 repetitions were performed, after identifying the best nucleotide substitution model. With regard to the fungal species, cycle sequencing with forward and reverse primers was performed using the Microseq D2 LSUrDNA fungal sequencing kit (Thermo Fisher Scientific, Waltham, MA, USA). Subsequently, the samples were analyzed using the 3500 Genetic Analyzers (Applied biosystems, Waltham, MA, USA). Species identification was performed by comparing the obtained sequences with those available in GenBank.

#### 2.2.4. DNA Extraction and PCR Conditions for Bovine Herpesvirus 1 (BoHV-1)

DNA was extracted using the ExtractMe genomic DNA kit (Blirt, Gdańsk, Poland) following the manufacturer’s instructions. The detection of BoHV-1 was performed using real-time PCR with the CFX96 Touch™ Real-Time PCR Detection System (Bio-Rad, Hercules, CA, USA). The Platinum™ qPCR SuperMix-UDG Kit (Invitrogen, Waltham, MA, USA) was used for the reaction mix, along with primers and TaqMan probe manufactured by Thermo Fisher Scientific and specifically designed in a previous study [21]. The reaction volume of 25 μL consisted of 5 μL of template, 180 nM of forward primer (5′-TGTGGACCTAAACCTCACGGT-3′), 180 nM of reverse primer (5′-GTAGTCGAGCAGACCCGTGTC-3′), 120 nM of TaqMan Probe (5′-FAM-AGGACCGCGAGTTCTTGCCGC-TAMRA-3′), 12,5 μL of Platinum™ Quantitative PCR SuperMIx-UDG, and nuclease-free water up to the final volume. The assay was carried out using the following PCR cycling conditions: one cycle at 50 °C for 2 min, one cycle at 95 °C for 2 min followed by 45 cycles of denaturation at 95 °C for 15 s and annealing/extension at 60 °C for 45 s. Positive control for BoHV-1 (virus cultured on MDBK cell line) as well a negative control (distilled water) were added into our molecular analysis.

#### 2.2.5. RNA Extraction and PCR Conditions for Bovine Parainfluenza Virus 3 (BPIV-3)

RNA was extracted using the QIAamp^®^ Viral RNA Mini Kit (Qiagen, Hilden, Germany) following the manufacturer’s protocol. The detection of BPIV-3 was performed using real-time PCR with the CFX96 Touch™ Real-Time PCR Detection System (Bio-Rad, Hercules, CA, USA). The SuperScript™ III Platinum™ One-Step qRT-PCR Kit (Invitrogen, Waltham, MA, USA) was used for the reaction mix, along with primers and TaqMan probe manufactured by Thermo Fisher Scientific. The amplification of BRSV was performed according to Horwood et al. [22]. The reaction volume of 20 μL consisted of 2 μL of template, 200 nM of forward primer (5′-TGTCTTCCACTAGATAGAGGGATAAAATT-3′), 200 nM of reverse primer (5′-GCAATGATAACAATGCCATGGA-3′), 200 nM of TaqMan Probe (5′-VIC-TGCYAYGTGGACGAGGGCATGC-MGB-NFQ-3′), 0,4 μL of Platinum™ taq mix, 10 μL of 2× reaction mix, 1 μL of MgSO_4,_ and nuclease-free water up to the final volume. The assay was carried out using the following PCR cycling conditions: one cycle of reverse transcription at 50 °C for 15 min, one cycle of PCR initial activation step at 95 °C for 2 min followed by 45 cycles of denaturation at 95 °C for 15 s and annealing/extension at 60 °C for 1 min. Positive control for BPIV-3 (virus cultured on MDBK cell line) as well a negative control (distilled water) were added into our molecular analysis.

#### 2.2.6. RNA Extraction and PCR Conditions for Bovine Viral Diarrhea Virus (BVDV)

Viral RNA extraction was performed on 50 mg of lung tissue for the detection of bovine viral diarrhea virus (BVDV). Samples were homogenized with the addition of 750 µL of TRI Reagent (Sigma-Aldrich, Darmstadt, Germany) and incubated for 10 min at room temperature. Subsequently, 200 µL of chloroform was added and samples were incubated for an additional 15 min at room temperature before being centrifuged at 12,000× *g* for 15 min at 4 °C. The supernatants were collected and mixed with 500 µL of isopropanol before being incubated for 10 min at room temperature and centrifuged again at 12,000× *g* for 15 min at 4 °C. The supernatant was removed, and the pellets were then resuspended in 1 mL of 70.0% ethanol and centrifuged at 7500× *g* for 5 min at 4 °C. The pellets were then dried for 10 min and resuspended in 50 µL of RNase-free water before being stored at −80 °C.

The detection of BVDV was performed using real-time PCR with the CFX96 Touch™ Real-Time PCR Detection System (Bio-Rad, Hercules, CA, USA). The SuperScript™ III Platinum™ One-Step qRT-PCR Kit (Invitrogen, Waltham, MA, USA) was used for the reaction mix, along with primers and TaqMan probe manufactured by Thermo Fisher Scientific and specifically designed in previous studies [23,24]. The reaction volume of 25 μL consisted of 5 μL of template, 800 nM of forward primer (5′-GRAGTCGTCARTGGTTCGAC-3′), 800 nM of reverse primer (5′-TCAACTCCATGTGCCATGTAC-3′), 120 nM of TaqMan Probe (5′-FAM-TGCYAYGTGGACGAGGGCATGC-TAMRA-3′), 0,5 μL of Platinum™ taq mix, 12.5 μL of 2× reaction mix, and nuclease-free water up to the final volume. The assay was carried out using the following PCR cycling conditions: one cycle of reverse transcription at 48 °C for 10 min, one cycle of PCR initial activation step at 95 °C for 10 min followed by 45 cycles of denaturation at 95 °C for 15 s and annealing/extension at 60 °C for 1 min. Positive control for BVDV (virus cultured on MDBK cell line) as well a negative control (distilled water) were added into our molecular analysis.

#### 2.2.7. RNA Extraction and Real-Time PCR Conditions for Bovine Respiratory Syncytial Virus (BRSV)

Viral RNA extraction was performed on 50 mg of lung tissue for the detection of bovine respiratory syncytial virus (BRSV). Samples were homogenized with the addition of 750 µL of TRI Reagent (Sigma-Aldrich, Darmstadt, Germany) and incubated for 10 min at room temperature. Subsequently, 200 µL of chloroform was added and samples were incubated for an additional 15 min at room temperature before being centrifuged at 12,000× *g* for 15 min at 4 °C. The supernatants were collected and mixed with 500 µL of isopropanol before being incubated for 10 min at room temperature and centrifuged again at 12,000× *g* for 15 min at 4 °C. The supernatant was removed, and the pellets were then resuspended in 1 mL of 70.0% ethanol and centrifuged at 7500× *g* for 5 min at 4 °C. The pellets were then dried for 10 min and resuspended in 50 µL of RNase-free water before being stored at −80 °C.

The detection of BRSV was performed using real-time PCR with the CFX96 Touch™ Real-Time PCR Detection System (Bio-Rad, Hercules, CA, USA). The SuperScript™ III Platinum™ One-Step qRT-PCR Kit (Invitrogen, Waltham, MA, USA) was used for the reaction mix, along with primers and TaqMan probe manufactured by Thermo Fisher Scientific. The amplification of BRSV was performed according to Boxus et al. [25]. The reaction volume of 25 μL consisted of 5 μL of template, 100 nM of forward primer (5′-GRAGTCGTCARTGGTTCGAC-3′), 100 nM of reverse primer (5′-TCAACTCCATGTGCCATGTAC-3′), 200 nM of TaqMan Probe (5′-FAM-TGCYAYGTGGACGAGGGCATGC-TAMRA-3′), 0,3 μL of Platinum™ taq mix, 12.5 μL of 2× reaction mix, and nuclease-free water up to the final volume. The assay was carried out using the following PCR cycling conditions: one cycle of reverse transcription at 50 °C for 15 min, one cycle of PCR initial activation step at 95 °C for 2 min followed by 45 cycles of denaturation at 95 °C for 15 s and annealing/extension at 60 °C for 1 min. Positive control for BRSV (virus cultured on MDBK cell line) as well a negative control (distilled water) were added into our molecular analysis.

### 2.3. Histopathological Investigations

Systemic post-mortem examination was performed on the cattle, and lungs and tracheas were grossly evaluated. All the lungs also underwent histopathological investigation in order to confirm macroscopic diagnosis. Samples were collected and fixed in a 10.0% neutral buffered formalin solution. They were routinely embedded in paraffin wax blocks, sectioned at 5 μm thickness, mounted on glass slides, and stained with Haematoxylin and Eosin. The samples were observed by means of light microscopy Axio Scope A1 (ZEISS, Oberkochen, Germany).

### 2.4. Statistical Analysis

Proportional incidence was calculated for the species of *Mycoplasma* that were detected more frequently (*M. bovis* and *M. dispar*). The other *Mycoplasma* species were not included in the analyses due to their low sample size. Comparison of proportional incidences between season, sex, age group, and weigh group were performed by calculating proportional incidence ratios (PIRs) using Poisson regression with adjusted (robust) variances. Variables characterized by a *p*-value <  0.20 at the univariate analysis were taken forward for multivariable modeling. Variables to be included in the final models were selected using a backward stepwise approach: variables were removed one at a time from the full model and retained if the likelihood ratio test returned a level of significance <  0.05. Other variables not included in the analysis were discarded due to data sparsity. Statistical analysis was performed with Stata/SE 16 (StataCorp LLC, 2019).

## 3. Results

Among the 322 cattle analyzed, 54 animals (16.8%) tested positive for *Mycoplasma* spp. The distribution of positive cases across years was: 11 (20.4%) in 2020, 9 (16.7%) in 2021, 21 (38.9%) in 2022, and 13 (24.1%) from January to June 2023 (Table 1). The identification of the sequences by comparison via BLAST with those deposited in the database has allowed the identification of nine different species belonging to Mycoplasmataceae: *M. alkalescens*, *M. arginini*, *M. bovigenitalium*, *M. bovirhinis*, *M. bovis*, *M. canadense*, *M. dispar*, *M. hyopharyngis*, and *U. diversum* (Table 1). The 54 sequences obtained were then compared with those already available in the online database (Figure 2). *M. bovis* and *M. dispar* were the most frequently detected species among the examined cattle (*n* = 22, 40.7%, and *n* = 13, 24.1%, respectively). Additionally, *U. diversum* and *M. alkalescens* were identified in 6 (11.1%) and 4 (7.4%) cattle, respectively. Less frequently detected *Mycoplasma* species included *M. bovirhinis* (*n* = 3, 5.6%), *M. arginini* (*n* = 2, 3.7%), and *M. canadense* (*n* = 23.7%). Lastly, *M. bovigenitalium* and *M. hyopharyngis* were identified in single cases (1.9% each).

Following the creation of an alignment of the obtained sequences with some used as reference downloaded from the GenBank database, the analysis of the nucleotide substitution model for the construction of a phylogenetic tree identified the Tamura-3-parameter model in association with discrete gamma distributions as the best choice. Phylogenetic analysis of the previously identified sequences allowed us to confirm the diagnosis, as well as to highlight a certain degree of nucleotide differences within the clones belonging to the same *Mycoplasma* species (Figure 2). All sequences have been deposited in GenBank; the identifications, sequence Accession Numbers, and references of the reference strains used for the construction of the phylogenetic tree are available in Appendix A.

Of the 54 cattle that tested positive for *Mycoplasma* spp., 9 animals (16.7%) did not exhibit co-infection with other pathogens. Among these, *M. bovis* emerged as the most frequently detected species (*n* = 5, 55.6%), followed by *M. bovirhinis* (*n* = 2, 22.2%), while *M. dispar* and *M. alkalescens* were each identified in a single animal (11.1% each). Necropsy and histological examination revealed catarrhal bronchopneumonia in 5 cases (55.6%), associated with *M. bovis* (*n* = 4) and *M. dispar* (*n* = 1). Additionally, purulent catarrhal bronchopneumonia was observed in 3 cattle (33.3%), associated with *M. bovis* (*n* = 1) and *M. bovirhinis* (*n* = 2). Notably, the necropsy and histological examination of the bovine (animal ID 21637/23) that tested positive for *M. alkalescens* revealed fibrinopurulent arthrosynovitis, rather than pneumonia. This finding was consistent with its history of progressive chronic weight loss and weakness (Table 1).

The remaining 45 cattle (83.3%) harbored co-pathogens. The species most frequently detected was *M. bovis*, (*n* = 17, 37.8%), followed by *M. dispar*, (*n* = 12, 26.7%). Additionally, *U. diversum* and *M. alkalescens* were detected in 6 (13.3%) and 3 (6.7%) cattle, respectively. Other less frequently detected *Mycoplasma* species included *M. canadense*, *M. arginini*, *M. bovirhinis*, *M. bovigenitalium*, and *M. hyopharyngis*. The necropsy and the histological examination of the 45 cattle revealed catarrhal bronchopneumonia in 17 animals (37.8%). In addition, purulent catarrhal bronchopneumonia (*n* = 11, 24.4%), bronchointerstitial pneumonia (*n* = 1, 2.2%), and necrosuppurative bronchopneumonia (*n* = 3, 6.7%) were noted. Finally, catarrhal fibrinopurulent bronchopneumonia (*n* = 4, 8.9%), fibrinonecrotizing hemorrhagic pneumonia (*n* = 4, 8.9%), and purulent bronchopneumonia (*n* = 4, 8.9%) were also observed. Bronchopneumonia coexisted with pleurisy in 12 cattle (26.7%) and with emphysema in 5 (11.1%), although no statistical association was demonstrated. Additionally, enteritis was frequently observed in cases where *Escherichia coli* was detected in the lung, but no statistical association could be established. Notably, the necropsy of a bovine (animal ID 18123/22) revealed fibrinopurulent pleurisy and chronic peritonitis, but no bronchopneumonia (Table 1).

Of the 45 cattle with co-infection, 30 (66.7%) harbored one additional pathogen, 10 (22.2%) had two additional pathogens, and 4 (8.9%) exhibited polymicrobial infection. In one instance (2.2%, sample 86346/21), the presence of 3 additional pathogens (*Pasteurella multocida*, BRSV, and BPIV-3) was observed (Table 1). The main bacterial pathogens identified in cattle exhibiting co-infection were *P. multiocida*, *E. coli*, *Mannheimia haemolytica*, *Trueperella pyogenes*, and *Histophilus somni* (Table 1). However, other bacterial and fungal species were also identified, but less frequently. These included *Corynebacterium kutscheri*, *Weissella cibaria*, *Acinetobacter schindleri*, *Clostridium perfringens*, *Proteus vulgaris*, and *Aspergillus fumigatus* (Table 1). Among co-infected cattle, *P. multocida* emerged as the most prevalent bacterial pathogen (*n* = 11, 24.4%). It was associated with *M. bovis* (*n* = 4), *M. dispar* (*n* = 3), *M. arginini* (*n* = 2), *M. alkascens* (*n* = 1), and *M. canadense* (*n* = 1). *E. coli* was detected in 11 cases of co-infection (24.4%) and was associated with *M. dispar* (*n* = 6), *M. bovis* (*n* = 3), and *U. diversum* (*n* = 2). *Mannheimia haemolytica* was identified in 4 cases of co-infection (8.9%) and was associated with *M. bovis* (*n* = 2), *M. alkascens* (*n* = 1), and *M. canadense* (*n* = 1). Lastly, *Histophilus somni* was detected in 3 cases of co-infection (6.7%), associated with *M. bovis* (*n* = 2) and *M. dispar* (*n* = 1).

The main viral pathogens identified in 45 cattle exhibiting co-infection were BRSV, BoHV-1, and BPIV-3 (Table 1), whereas the presence of BVDV was not observed. Specifically, BRSV was detected in 6 cattle (13.3%), and was associated with *M. bovis* (*n* = 4), *M. dispar* (*n* = 1), and *U. diversum* (*n* = 1). BoHV-1 was also detected in 6 cattle (13.3%) and was associated with *M. bovis* (*n* = 2), *M. alkascens* (*n* = 2), *M. canadense* (*n* = 1), and *U. diversum* (*n* = 1). BPIV-3 was instead detected in 3 cases of co-infection (6.7%), associated with different *Mycoplasma* species: *M. bovis*, *M. dispar*, and *U. diversum*.

No discernible pattern of etiological agent occurrence was observed within or across the 46 farms studied. This suggests that the distribution of etiological agents in these herds was largely random, with no clear evidence of clustering or predominance of specific agents within any considered farms.

Of the 54 cattle testing positive for *Mycoplasma* spp., 28 (51.9%) were younger than 5 months old. *M. bovis* (*n* = 11, 39.3%) and *M. dispar* (*n* = 11, 39.3%) were the predominant species in this age group, while other species such as *U. diversum*, *M. bovirhinis*, and *M. alkascens* were identified less frequently. The remaining 26 cattle (48.2%) were older than 5 months with only 6 animals exceeding 12 months (Table 1). *M. bovis* remained the most prevalent species (*n* = 11, 42.3%), followed by *M. alkalescens* (*n* = 3, 11.5%), *U. diversum* (*n* = 3, 11.5%), *M. bovirhinis* (*n* = 3, 11.5%), *M. dispar* (*n* = 2, 7.7%), and *M. canadense* (*n* = 2, 7.7%). *M. bovigenitalium* and *M. hyopharyngis* were each detected in one bovine (3.8% each).

Respiratory symptoms in the examined cattle were predominantly observed during the winter (*n* = 19, 35.2%), between January 2020 and June 2023. A total of 7 cases were reported in March, compared to 6 in both January and February. During the autumn (*n* = 14, 25.9%), 5 cases were observed in October, 5 in November, and 4 in December. A total of 14 cases (25.9%) were also identified during the spring, with 6 cases in April, 5 in May, and 3 in June. Finally, during the summer period (*n* = 7, 13.0%), 3 cases were reported in July, none in August, and 4 in September (Appendix A). In winter, the most prevalent species was *M. bovis*, (*n* = 7), followed by *M. dispar* (*n* = 3), *U. diversum* (*n* = 3), and *M. alkascens* (*n* = 3). In spring, *M. bovis* remained the most prevalent species (*n* = 7). This species was again the most prevalent in autumn (*n* = 7), followed by *M. dispar* (*n* = 6). In summer, *M. dispar* (*n* = 2) and *M. bovirhinis* (*n* = 2) were the most common species, followed by *M. bovis*, *M. alkascens*, and *U. diversum* detected in a single animal. Notably, *M. canadense* and *M. bovigenitalium* were detected in winter while *M. arginini* and *M. hyopharyngis* in spring.

At the univariable analysis for *M. bovis*, no variables presented a *p*-value < 0.20, and thus, none were taken forward to the multivariable analysis. With regard to *M. dispar,* age (*p* = 0.02) and weight (*p* = 0.13) were considered for multivariable analysis. In the final model, ≤5 months-old animals were approximately five times more likely (PIR = 5.1, 95% CI 1.2–21.2, *p* = 0.02) to be infected by *M. dispar* (Table 2).

## 4. Discussion

*Mycoplasma bovis* was the most prevalent *Mycoplasma* species, infecting 40.7% of the positive animals, either alone or in combination with other pathogens. In fact, *M. bovis* was the predominant pathogen in Southern Italian farms, accounting for over half of all positive lung samples as the unique pathogen [26]. In Sicily, *M. bovis* was isolated in 90.0% of pneumonia cases, while no *Mycoplasma* species were found in the lungs without lesions, used as negative controls [27]. *M. bovis* was the predominant species in pneumonic veal calves from Northwestern Italy, accounting for 25.0% of cases and consistently linked to inflammatory conditions [14]. In Northeastern Italy, *M. bovis* was the most common bacterium isolated in feedlot cattle with respiratory disease, representing approximately 12.0% of cases [28]. In another study from Northeastern Italy, *M. bovis* was consistently found to be among the prevalent species [3]. In Switzerland, a high prevalence of *M. bovis* within severe and chronic cases of BRD was observed, suggesting this species is under detected in routine testing [12,15]. In Australia, the majority of the feedlot cattle, receiving treatment for BRD, were positive for *M. bovis* (90.0%) alone or in association with other pathogens [29]. *M. bovis* was the most frequently identified in England, representing 32.0% of the *Mycoplasma* species detected in 4447 cattle. In addition, it was the unique detected cause of respiratory disease in 28.0% of the pneumonia cases and an important factor in the multifactorial bovine respiratory disease complex [2]. A previous study performed in England reported *M. bovis* as the most frequently detected pathogen among cattle examined between 1990 and 2000. The majority of these bacteria were isolated from the lungs or upper respiratory tract [30]. In France, *M. bovis* was frequently linked to respiratory disease in both unweaned and weaned calves. Moreover, it was consistently identified as the most common *Mycoplasma* species isolated from the respiratory tract of young animals diagnosed with BRD [31,32]. In Lithuania, only *M. bovis* was isolated from the lungs in 73.0% of cases of bronchointerstitial pneumonia while other microorganisms were not isolated [4]. Another study performed in Lithuania reported *M. bovis* as the only *Mycoplasma* species isolated from cattle lung samples, accounting for 60.0% of bronchointerstitial pneumonia cases. Additionally, it was the most frequently detected species in the nasal cavities of the examined cattle [33]. Therefore, our findings and those previously reported suggest that *M. bovis* can be recognized as one of the most pathogenic organisms involved in BRD, due to its association with lung inflammation.

*Mycoplasma dispar* was the second most frequently detected species in this study (24.1% of positives). In Northeastern Italy, *M. dispar* was less detected, accounting for 12.0% of the cases identified in 711 cattle. [3]. In England, *M. dispar* was frequently identified, representing 11.0% of the pneumonia diagnoses. Moreover, it was considered a cattle pathogen due to its association with respiratory disease and its ability to trigger both pneumonia in experimental infections and cytopathic effects on bovine cell line [2]. In Finland, *M. dispar* was the first species detected, found in 91.0% of the animals analyzed and representing the most common isolate from lavage fluid of affected calves [34]. These findings suggest that *M. dispar* may act as a pathogen in cattle, contributing to the development of pneumonia. Furthermore, in our study, this species was the only pathogen identified in the lung of a bovine. However, further investigations are required to definitively establish its pathogenicity, since *M. dispar* was also present in the respiratory tract of healthy cattle [2].

The proportion of positive animals infected with *Ureaplasma diversum* in this study (11.1%) is consistent with the data obtained in previous studies in Italian cattle (9.0%), based on the analysis of nasal swabs [3]. However, a lower proportion of positive animals (3.0%) was reported in lung tissue and rarely in the reproductive tract of English cattle. Therefore, its role in respiratory disease was considered opportunistic, rather than as a primary cause of BRD [2]. Another study considered *U. diversum* an opportunistic pathogen, as it was isolated from the nasal cavity of asymptomatic calves. However, the risk of infections appears to be higher in younger cows and is associated with the development of catarrhal pneumonia [1]. In Australia, *U. diversum* has been observed to be more prevalent in calves with clinical respiratory disease compared to their visually healthy counterparts, suggesting this species may contribute to the development of BRD [29]. In this study, *U. diversum* was always detected in association with other pathogens in the pneumonic lung of cattle, indicating its potential involvement in the pathogenesis of BRD as an opportunistic pathogen.

A bacteriological examination of 17 bronchoalveolar lavage samples from Danish calves with pneumonia identified *Mycoplasma alkascens* in 8 samples alone or in association with other pathogens [35]. In England, *M. alkascens* was detected in the lung of cattle diagnosed with BRD, accounting for 15.0% of the *Mycoplasma* species detected in 4447 animals [2]. This *Mycoplasma* species was also isolated twice from the lungs of English cattle and in cases of mastitis and arthritis [30]. In France, *M. alkascens* has emerged in cattle and could rapidly spread to other European countries, being involved in respiratory disease, arthritis, and mastitis [32]. In this study, *M. alkascens* was detected in 4 cattle (7.4% of positives), confirming it as an emerging species, not only involved in respiratory disease. In fact, a bovine (animal ID 21637/21) that only harbored *M. alkascens* in the lung did not exhibit pneumonia, but rather fibrinopurulent arthrosynovitis (Table 1). In contrast, the other 3 cattle harbored other pathogens in the lung along with *M. alkascens*. This suggests that it may act either as a secondary invader or as an opportunistic pathogen, even if it was also detected alone in the bronchoalveolar lavage samples from calves with pneumonia, acting as a pathogen [35]. Therefore, further investigations are required to clarify its pathogenicity.

In Northeastern Italy, *Mycoplasma bovirhinis* was identified in 40.0% of nasal swabs, with a higher prevalence observed during warmer seasons [3]. In Egypt, 6.0% of the cattle of various ages tested positive for *Mycoplasma* spp., including two isolates of *M. bovirhinis* [6]. In England, *M. bovirhinis* accounted for 20.0% of the *Mycoplasma* species identified in 4447 animals [2]. In Denmark, the bacteriological analysis of 17 bronchoalveolar lavage samples from calves with pneumonia identified *M. bovirhinis* in 2 samples [35]. In Poland, 4.0% of cattle exhibiting clinical signs suggestive of *Mycoplasma* infection tested positive for *M. bovirhinis* [16]. In Finland, *M. bovirhinis* was found to be equally prevalent in both healthy and diseased cattle, consistent with findings from another study [36,37]. In this study, *M. bovirhinis* was detected in the lungs of 3 cattle (5.6% of positives). Notably, it was the sole pathogen identified in the pneumonic lungs of 2 animals, which aligns with findings from other studies [35,36]. This suggests that *M. bovirhinis* could be considered a pathogenic species, contributing to the development of pneumonia. However, *M. bovirhinis* was also detected in the lungs of clinically healthy cattle, even if its detection rate in such animals appears to be lower than that in cattle with pneumonia [3,38]. Therefore, its pathogenicity has not been clearly established, and further investigations are required.

*Mycoplasma arginini* accounted for 6.0% of *Mycoplasma* species identified in 4447 English cattle [2]. In the USA, *M. arginini* has been isolated from the respiratory tracts and milk samples of ruminant livestock, particularly in farms affected by mastitis [17]. In Egypt, 460 lung samples from camels were randomly collected at the abattoir. A total of 210 samples were pneumonic, with 48 lungs (22.9%) tested positive for *M. arginini* [39]. Another Egyptian study reported that *Mycoplasma* species were isolated from 5 healthy cattle lungs, with 3 of them being *M. arginini*. Additionally, 14 *Mycoplasma*s were isolated from pneumonic lung samples, with 3 of them identified as *M. arginini* [36]. In France, *M. arginini* was frequently isolated, both alone and in association with other pathogens, although its diagnostic significance remains limited [32]. Other studies have reported that *M. arginini* was frequently detected in association with other pathogens and can be isolated from healthy bovine lungs [36,40]. In our study, this species was detected twice in the lung of cattle (3.7% of positives) and always in association with other pathogens. These findings suggest that *M. arginini* may act as an opportunistic pathogen or secondary invader, although it could also induce a pulmonary infection by itself [36,39].

*Mycoplasma canadense* is usually associated with mastitis and the infection of the reproductive system [3]. In fact, this species was isolated from the outbreaks of vulvovaginitis in Israelian dairy herds and was considered as probable pathogens for bovine genital tract disorders [41]. In the USA, *M. canadense* has been primarily isolated from quarter and pooled milk samples, suggesting a potential role in mammary gland health. Among the 889 *Mycoplasma* isolates, 5.8% were identified as *M. canadense* [42]. With regard to the respiratory tract, *M. canadense* accounted for 2.0% of *Mycoplasma* species identified in 4447 English cattle [2]. In Southern Italy, lung tissue samples from 104 cattle revealed lung lesions in 77 animals, with 11 samples tested positive for *Mycoplasma* spp. This species was identified in a lung that showed lesions of fibrinous pleuropneumonia without the presence of other pathogens [27]. In this study, *M. canadense* was detected twice in bovine lungs (3.7% of positives), always in association with other pathogens. Although *M. canadense* is recognized as a probable pathogen in bovine genital diseases, its frequent co-occurrence with other pathogens suggests a possible role as an opportunistic pathogen or secondary invader, in the respiratory tract.

*Mycoplasma bovigenitalum* is usually associated with mastitis and disorders of the reproductive tract [3,6]. In fact, this species was isolated from the outbreaks of vulvovaginitis in Israelian dairy herds along with *M. canadense* [41]. In the USA, among the 889 *Mycoplasma* isolates obtained from milk samples, 6.5% were identified as *M. bovigenitalum* [42]. Other studies reported that *M. bovigenitalum* has a pathogenic impact on the reproductive system, suggesting it could be considered a probable pathogen for the genital tract of cattle [3,6,8]. On the other hand, *M. bovigenitalum* has been isolated from 5 pneumonic lungs of Egyptian cattle [36]. *M. bovis* was the predominating *Mycoplasma* species isolated from pneumonic lungs (30.0%) followed by *M. bovigenitalum* (17.0%). The lungs of mice inoculated with these two species exhibited congestion compared to those infected with *M. arginini* [36]. In England, *M. bovigenitalum* accounted for 1.0% of *Mycoplasma* species identified in 4447 cattle. It was most often identified in vaginal samples of adult animals followed by the lungs of younger cattle below 5 months [2]. Our study identified *M. bovigenitalum* in only one bovine (1.9% of positives), consistent with findings from English studies. The identification of a polymicrobial infection within the same lung suggests that *M. bovigenitalum* may be involved in the progression of BRD as an opportunistic pathogen. However, further research is necessary to definitively understand its pathogenicity in the lower respiratory tract.

*Mycoplasma hyopharyngis,* a member of the *M. fermentans* group, is most closely related to *M. hyosynoviae* [43]. Initially isolated from the nasal and pharyngeal mucosae of clinically normal swine, its pathogenicity remains uncertain [40]. However, its recovery from nasal secretions of a pig in a large confinement herd in the USA suggests a potential role in respiratory disease [44]. Serotypically distinct from other *Mycoplasma* species, *M. hyopharyngis* has been isolated from the pharyngeal and nasal cavities of pigs [44]. Notably, it has also been detected in a bovine digital dermatitis lesion in South Korea and isolated from a lame pig’s joint in Hungary [45,46]. This study, for the first time, reports the detection of *M. hyopharyngis* in the pneumonic lung of a bovine with BRD in association with *Proteus vulgaris*. This finding suggests the potential for *M. hyopharyngis* to infect cattle lungs, although further research is necessary to elucidate its full host range and pathogenic potential.

The major limitation of our study in the identification of *Mycoplasma* species is represented by the diagnostic technique used. In fact, although the PCR-sequencing protocol adopted was effective in identifying infections caused by a single *Mycoplasma* species, it was not possible to identify the causative agents in the cases of suspected co-infections. The electropherograms obtained in these cases were in fact not legible, and therefore were excluded from the present study. Our study therefore aims to be a first step to understand the species most circulating in cattle farms of the Piedmont region; future studies should focus on the identification of the *Mycoplasma* species causing co-infections, implementing the diagnostic protocol by means of species-specific PCR protocols, or using techniques that allow the simultaneous identification of multiple species (e.g., 16S metabarcoding).

Respiratory diseases caused by *Mycoplasma* spp. exhibit nonspecific clinical signs, often resembling infections caused by other pathogens of the bovine respiratory tract, especially in cases of co-infection [14]. In Switzerland, a histopathological examination of 104 pneumonic lungs, revealed 79 cases of bronchopneumonia, 16 of bronchointerstitial pneumonia, and 9 of interstitial pneumonia. In all these cases, the infection with BPVI-3, *M. bovis*, and others *Mycoplasma* species was considered the cause of pneumonia [12]. In Lithuania, *M. bovis* was isolated in 72.7% of cases diagnosed with bronchointerstitial pneumonia. No other microorganisms were isolated from the pneumonic lungs, which also exhibited signs of emphysema [4]. Another study reported that the histological examination of cattle lungs, at the time of slaughter, showed signs of pneumonia, typical of chronic bronchointersitial pneumonia, in 14.3% of the samples [33]. In Poland, *M. bovis* was isolated from caseonecrotic and fibrinosuppurative bronchopneumonia [8]. In Canada, *M. bovis* was isolated from 98.0% of cases with caseonecrotic bronchopneumonia, and in one case, fibrinosuppurative bronchopneumonia was exclusively associated with *M. bovis*. *M. arginini*, *U. diversum*, and *M. bovirhinis* were less commonly isolated from caseonecrotic bronchopneumonia [47]. In Finland, *M. dispar* and *Pasteurella multocida* together were isolated in 13 tracheobronchial fluid samples of young cattle. The post-mortem examination of these animals revealed signs of chronic fibrinopurulent bronchopneumonia [34]. In Brazil, 91.4% of cattle diagnosed with BRD exhibited clinical signs consistent with pneumonia. Interstitial pneumonia (46.8%) was the most predominant pattern of pulmonary disease observed, followed by necrosuppurative bronchopneumonia (28.1%) and suppurative bronchopneumonia (18.7%). In Northwestern Italy, *M. bovis* was the most common finding in calves being isolated from 25.0% of pneumonic cases and was consistently associated with inflammatory findings [14]. Lung lesions associated with *M. bovis* mainly developed into a severe necrosuppurative bronchopneumonia or fibrinonecrotizing pneumonia. Conversely, a slight association between *M. bovis* with catarrhal broncopneumonia and bronchointerstitial pneumonia was observed [14]. In Southern Italy, *M. bovis* was isolated in 90.0% of pneumonia cases and in 83.0% of cases of fibrinous pleuropneumonia. Histopathological lesions were observed in all samples that tested positive for *Mycoplasma* spp. These lesions were represented by catarrhal or catarrhal-purulent bronchopneumonia in 4 cattle, while the remaining showed findings of fibrinous pleuropneumonia [27]. In this study, necropsy and histological examination of 9 animals without co-infection predominantly revealed catarrhal bronchopneumonia. Similarly, examinations of 45 cattle with co-infections predominantly revealed catarrhal bronchopneumonia or purulent catarrhal bronchopneumonia. The histopathological changes observed in co-infections can indicate the disease process and suggest the likely types of pathogens involved, although more specific assays are necessary to identify all pathogens present. Our findings are in agreement with previous reports, even if the presence of catarrhal bronchopneumonia was less frequently observed [8,14,47]. Interestingly, catarrhal and purulent bronchopneumonia were linked with emphysema in five cattle, a finding consistent with observations reported by Gabinaitiene et al. [4]. Furthermore, the frequent association of enteritis with *Escherichia coli* detection in the lung suggests that this bacterium may initially induce gastroenteric inflammation, subsequently migrating to the lung as a secondary invader. 

In this study, *M. bovis* has been reported to cause pneumonia in cattle and *P. multocida* emerged as the most frequent bacterial pathogen among co-infected animals, followed by *Escherichia coli*, *Mannheimia haemolytica*, *and Histophilus somni*. With regard to viral pathogens, BRSV and BoHV-1 were the most frequently detected viral pathogens in cases of co-infection, followed by BPIV-3. An association with pneumonia cases and *M. bovis* along with Pasteurellaceae family pathogens, especially for *M. haemolytica* and *H. somni*, has been demonstrated [8,11,26,47]. In Switzerland, the most common identified microorganisms in association with *M. bovis* were *P. multocida*, *M. haemolytica*, *H. somni*, BRSV, BoHV-1, BVDV, and BPIV-3 [5]. Furthermore, an examination of 104 lungs of cattle, revealed co-infections with BPIV-3 and *Mycoplasma* spp. in approximately half of the cases [12]. In Denmark, *H. somni*, *M. haemolytica*, and *P. multocida* were the bacteria most associated with bronchopneumonia in calves [35]. In Canada, *P. multocida* was the most frequent bacterium isolated in cattle diagnosed with BRD (54.8%), followed by *M. haemolytica* (30.5%) and *Histophilus somni* (22.9%), in association with *M. bovis* [7]. In Finland, the most common isolate from lavage fluid of diseased calves was *M. dispar* in association with *P. multocida* [34]. In Ireland, *M. bovis* was isolated in 18% of cattle that died from pneumonia. Other respiratory pathogens such as *P. multocida*, *M. haemolytica*, BPIV-3, and BoHV-1 were identified in 66% of the *Mycoplasma bovis*-positive cases [48]. In Northwestern Italy, *M. bov*is was commonly isolated from bronchopneumonic lesions of cattle diagnosed with BRD often in association with other bacterial pathogens such as *M. haemolytica*, *P. multocida*, and *H. somni* [14]. In Brazil, BVDV (56.2%) was the most frequently identified pathogen, followed by *M. bovis* (50.0%), BoHV-1 (43.7%), BRSV (34.4%), and BPIV-3 (15.6%). Viral infections were identified in 50% of the cases without any association with other pathogens, while *M. bovis* was identified in the remaining cases [49].

In this study, 28 cattle that tested positive for *Mycoplasma* spp. were less than 5 months old (51.9%). Of the remaining 26 cattle (48.1%), all were older than 5 months of age, and only 6 (11.1%) were 24 months or older. Cattle under 5 months of age were approximately 5 times more likely to be infected with *M. dispar* than those older than 5 months. This suggests that the maturation of the immune system in cattle may contribute to increased resistance against various pathogens, including *M. dispar*. Furthermore, older cattle may have previously encountered *M. dispar*, leading to the development of a certain level of immunity through prior exposures. No significant differences in proportional incidences were observed for other *Mycoplasma* species between the two age groups. However, this hypothesis requires further verification with an appropriate control group and this study may have been limited by the small number of positive cases included in the analyses. Nonetheless, *Mycoplasma* infections appear to be more frequent in calves and young cattle, whereas adult animals are rarely affected. Our findings are consistent with those already reported in other studies. In fact, the prevalence of *M. bovis* varied significantly with age in Lithuania. It was more frequently isolated from the upper respiratory tract of the cattle less than 3 months old compared to those aged 17 months [33]. In Northwestern Italy, *M. bovis* was the most common bacterium isolated in calves diagnosed with pneumonia (25.0%) and was consistently associated with inflammatory findings [14]. In Poland, 87.0% of the examined calves exhibited clinical signs of BRD, while a lower prevalence of 21.2% in adult cattle was observed. The most frequently identified *Mycoplasma* species was *M. bovis* [16]. In France, *M. bovis* was commonly associated with respiratory disease in unweaned and weaned calves but has rarely been identified in dairy cattle [31]. In Switzerland, *M. bovis* was observed to cause pneumonia in very young calves, while the pneumonic lesions in the lungs of adult animals were detected only in cattle with viral or bacterial co-infections [5]. Similarly, the presence of multiple respiratory pathogens was observed in 75.0% of adult dairy cattle diagnosed with pneumonia in Brazil, suggesting that co-infections may be more prevalent in adult animals [49].

In this study, the majority of respiratory symptoms in the cattle examined were observed during the winter period (19 cases, 35.2%). A total of 14 (25.9%) cases were detected in both autumn and spring, compared to only 7 (13.0%) cases reported in the summer. Cattle that tested positive for *Mycoplasma* spp. did not show a significant difference in proportional incidences of *M. bovis* infection between the warm and the cold season. Other studies investigated the effect of weather conditions. For instance, a seasonal variation in the prevalence of *M. bovirhinis* was observed in Northeastern Italy, with a notable decrease during the winter months compared to the warmer seasons [3]. In Algeria, *M. bovis* was consistently detected in 64% of the calves during winter and spring, but the prevalence was significantly lower during the summer and autumn months [13]. Conversely, in Southern Italy, BRD incidence remained relatively stable throughout the study period, exhibiting no significant correlation with colder months [26].

The results of this study provide valuable insights into the prevalence and impact of various *Mycoplasma* species on BRD in cattle. These findings can be applied to develop more effective management practices and emphasize the importance of implementing early detection programs, regular herd screening, and targeted testing of high-risk animals or herds. Therefore, this research may contribute to mitigating the impact of *Mycoplasma* species on BRD and ultimately to improving the overall health and productivity of cattle herds.

## 5. Conclusions

*Mycoplasma bovis* was identified as one of the most frequent pathogens in respiratory diseases in cattle in Northwestern Italy. Its association with severe and chronic BRD cases underscores its significance as a primary pathogen. While *M. dispar* and *U. diversum* were also frequently identified, their roles as pathogens remain less clear, with evidence suggesting both primary and opportunistic involvement. Other *Mycoplasma* species, such as *M. alkascens, M. bovirhinis, M. arginini, M. canadense, and M. bovigenitalum*, were detected at lower frequencies but may still contribute to BRD. When *Mycoplasma* species are involved in co-infections with other pathogens, such as Pasteurellaceae family members or viruses, the lesions can be more severe and complex. In some cases, co-infections can lead to bronchitis, interstitial fibrosis, and emphysema. Histopathological examination revealed that bronchopneumonia was the most common lesion associated with *Mycoplasma* infections and the lung lesions can vary depending on the species of *Mycoplasma* involved and the presence of co-infections. The effect of age on the proportion of animals positive to *M. dispar* detected in this study highlights the necessity for further investigations into the potential role of age as a risk factor for *Mycoplasma* infections. Notably, this study identified *Mycoplasma hyopharyngis* in bovine lungs for the first time. Traditionally, *M. hyopharyngis* has been isolated from the respiratory tract of pigs. These findings suggest a potential role for *M. hyopharyngis* in the development of BRD and could impact current practices in cattle health management, as the transmission of *Mycoplasma hyopharyngis* could occur between cattle and swine. However, further studies, including the isolation and culturing of the strain, are useful to definitively confirm its role in BRD.

## Figures and Tables

**Figure 1 microorganisms-12-02340-f001:**
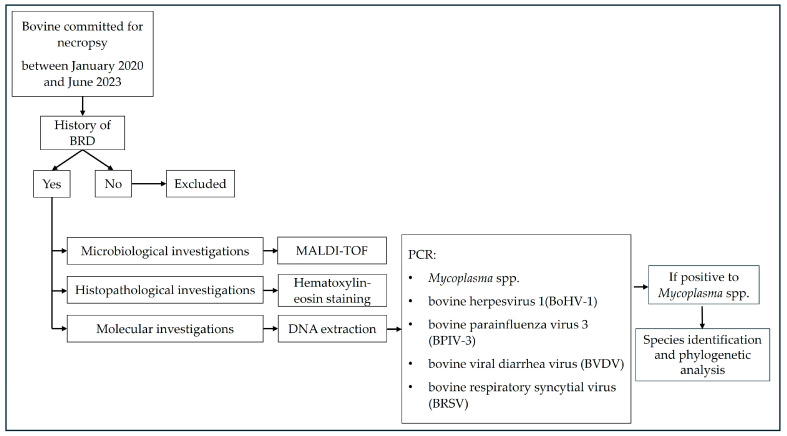
A flowchart of the experimental design.

**Figure 2 microorganisms-12-02340-f002:**
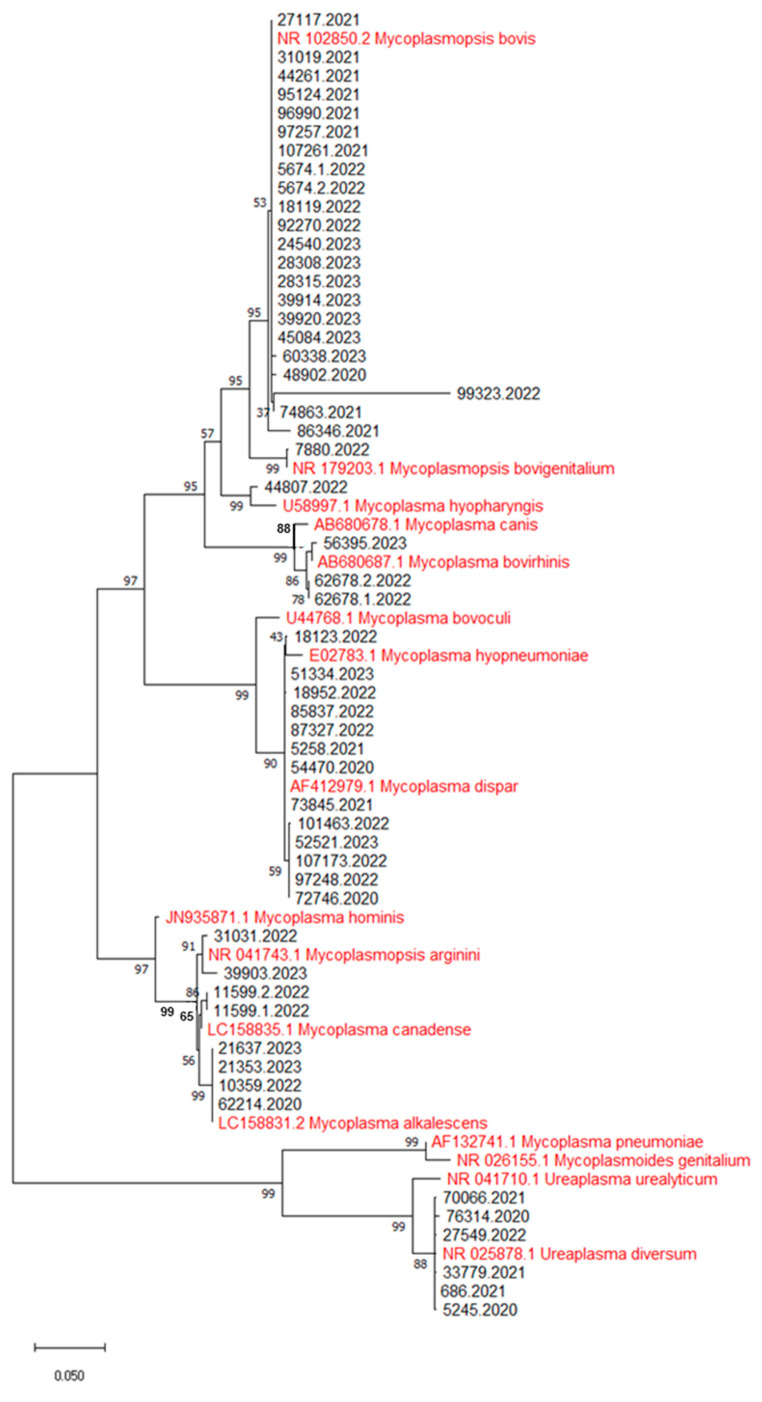
A phylogenetic tree constructed by including the 54 *Mycoplasma* sequences of this study and the reference sequences available in GenBank. The sequences of the present study are shown in black; those taken from GenBank are highlighted in red.

**Table 1 microorganisms-12-02340-t001:** Summary of cattle testing positive for various *Mycoplasma* species, including co-infection data and histopathological results.

Animal ID	Biological Matrix	Mycoplasma Species	Other Pathogens	Histopathological Results
48902/20	Lung	*Mycoplasma bovis*		Severe and diffuse catarrhal bronchopneumonia
76314/20	Lung	*Ureaplasma diversum*	BPIV-3	Diffuse catarrhal bronchopneumonia
62214/20	Lung	*Mycoplasma alkalescens*	*Mannheimia haemolytica*	Severe fibrinonecrotizing hemorrhagic pneumonia
50470/20	Lung	*Mycoplasma dispar*	*Pasteurella multocida*	Severe and chronic bronchointerstitial pneumonia
5245/20	Lung	*Ureaplasma diversum*	*Escherichia coli*	Severe fibrinopurulent catarrhal bronchopneumonia and enteritis
72746/20	Lung	*Mycoplasma dispar*	*Corynebacterium kutscheri*	Severe catarrhal bronchopneumonia
44261/21	Lung	*Mycoplasma bovis*		Acute and diffuse catarrhal bronchopneumonia and foamy trachea
686/21	Lung	*Ureaplasma diversum*	BoHV-1	Severe fibrinopurulent catarrhal bronchopneumonia
96990/21	Lung	*Mycoplasma bovis*	*Trueperella pyogenes*	Severe and subchronic fibrinopurulent catarrhal bronchopneumonia
33779/21	Lung	*Ureaplasma diversum*	BRSV and *Aspergillus fumigatus*	Severe fibrinopurulent catarrhal bronchopneumonia and pleurisy
74863/21	Lung	*Mycoplasma bovis*	*Pasteurella multocida*	Severe catarrhal bronchopneumonia
97257/21	Lung	*Mycoplasma bovis*	polymicrobism	Severe catarrhal bronchopneumonia
70066/21	Lung	*Ureaplasma diversum*	*Escherichia. Coli* and *Acinetobacter schindleri*	Severe catarrhal bronchopneumonia
107261/ 21	Lung	*Mycoplasma bovis*	*Clostridium perfringens*	Chronic catarrhal bronchopneumonia, necrotic hemorrhagic enterocolitis, and clostridial enterotoxemia
31019/21	Lung	*Mycoplasma bovis*	BRSV and *Pasteurella multocida*	Severe bilateral purulent bronchopneumonia
95124/21	Lung	*Mycoplasma bovis*	*Histophilus somni*	Severe and diffuse catarrhal bronchopneumonia and hemorrhagic tracheitis
73845/21	Lung	*Mycoplasma dispar*	*Escherichia coli and Aspergillus fumigatus*	Severe and chronic bronchointerstitial pneumonia
5258/21	Lung	*Mycoplasma dispar*	*Escherichia coli*	Severe fibrinonecrotizing hemorrhagic pneumonia and enteritis
27117/21	Lung	*Mycoplasma bovis*		Severe catarrhal bronchopneumonia
86346/21	Lung	*Mycoplasma bovis*	*Pasteurella multocida*, BRSV and BPIV-3	Diffuse catarrhal bronchopneumonia
11599.1/22	Lung	*Mycoplasma canadense*	*Trueperella pyogenes* and *BoHV-1*	Subchronic purulent catarrhal bronchopneumonia
11599.2/22	Lung	*Mycoplasma canadense*	*Mannheimia haemolytica*	Severe and chronic purulent bronchopneumonia
10395/22	Lung and trachea	*Mycoplasma alkalescens*	BoHV-1	Moderate bronchointerstitial pneumonia and a severe and diffuse fibrinopurulent and hemorrhagic tracheitis
18123/22	Lung	*Mycoplasma dispar*	*Escherichia coli*	Chronic fibrinopurulent pleurisy and chronic peritonitis
27549/22	Lung	*Ureaplasma diversum*	*Pasteurella multocida*	Severe catarrhal bronchopneumonia with a subcutaneous emphysema and pleurisy
92270/22	Lung	*Mycoplasma bovis*		Severe and diffuse purulent catarrhal bronchopneumonia with a subpleural emphysema and pleurisy
5674.1/ 22	Lung	*Mycoplasma bovis*	BoHV-1	Purulent catarrhal bronchopneumonia and subchronic pleurisy
5674.2/ 22	Lung	*Mycoplasma bovis*	BoHV-1	Severe purulent catarrhal bronchopneumonia
62678.1/22	Lung and trachea	*Mycoplasma bovirhinis*		Severe and diffuse purulent catarrhal bronchopneumonia
62678.2/22	Lung and trachea	*Mycoplasma bovirhinis*		Severe and diffuse purulent catarrhal bronchopneumonia
18119/22	Lung	*Mycoplasma bovis*	*Mannheimia haemolytica* and *Escherichia coli*	Severe purulent catarrhal bronchopneumonia and peritonitis
18952/22	Lung	*Mycoplasma dispar*	*Escherichia coli*	Diffuse catarrhal bronchopneumonia and enteritis
85837/22	Lung	*Mycoplasma dispar*	*Escherichia coli*	Acute catarrhal bronchopneumonia and enteritis
44807/22	Lung	*Mycoplasma hyopharyngis*	*Proteus vulgaris*	Severe purulent catarrhal bronchopneumonia
107173/ 22	Lung	*Mycoplasma dispar*	*Escherichia coli* and BRSV	Severe and diffuse purulent catarrhal bronchopneumonia and enteritis
97248/22	Lung	*Mycoplasma dispar*	*Pasteurella multocida*	Severe and subacute catarrhal bronchopneumonia
31031/22	Lung	*Mycoplasma arginini*	*Pasteurella multocida*	Severe subchronic purulent catarrhal bronchopneumonia and pleurisy
87327/22	Lung	*Mycoplasma dispar*	*Pasteurella multocida* and BPIV-3	Ssevere purulent catarrhal bronchopneumonia and pleurisy
99323/22	Lung	*Mycoplasma bovis*	*Mannheimia haemolytica*	Fibrinonecrotizing hemorrhagic pneumonia
101463/ 22	Lung	*Mycoplasma dispar*		Severe catarrhal bronchopneumonia
7880/22	Lung	*Mycoplasma bovigenitalium*	polymicrobism	Fibrinonecrotizing hemorrhagic pneumonia and pleurisy
21353/23	Lung and trachea	*Mycoplasma alkalescens*	*Pasteurella multocida* and BoHV-1	Purulent bronchopneumonia and congested trachea
24540/23	Lung	*Mycoplasma bovis*	*Escherichia coli* and BRSV	Chronic purulent bronchopneumonia, and emphysema
28308/23	Lung	*Mycoplasma bovis*	BRSV	Chronic catarrhal bronchopneumonia
28315/23	Lung	*Mycoplasma bovis*	*Trueperella pyogenes*	Chronic catarrhal bronchopneumonia
56395/23	Lung and trachea	*Mycoplasma bovirhinis*	polymicrobism	Congested trachea with foam, emphysema, pleurisy, and catarrhal bronchopneumonia
60338/23	Lung	*Mycoplasma bovis*	*Escherichia coli*	Necrosuppurative bronchopneumonia, fibrinopurulent pleurisy, and enteritis
21637/23	Lung	*Mycoplasma alkalescens*		Fibrinopurulent arthrosynovitis
39903/23	Lung	*Mycoplasma arginini*	*Pasteurella multocida*	Purulent catarrhal bronchopneumonia and pleurisy
39914/23	Lung	*Mycoplasma bovis*	*Pasteurella multocida*	Purulent catarrhal bronchopneumonia and pleurisy
39920/23	Lung and Pulmonary exudate	*Mycoplasma bovis*		Catarrhal bronchopneumonia and polyarthritis
45084/23	Lung	*Mycoplasma bovis*	*Histophilus somni*	Severe and diffuse catarrhal bronchopneumonia and subpleural emphysema
51334/23	Lung	*Mycoplasma dispar*	*Weissella cibaria*	Subchronic catarrhal bronchopneumonia
52521/23	Lung	*Mycoplasma dispar*	*Histophilus somni*	Chronic purulent catarrhal bronchopneumonia and pleurisy

**Table 2 microorganisms-12-02340-t002:** Proportional incidence ratios for *Mycoplasma bovis* and *Mycoplasma dispar*. PI = proportional incidence; PIR = proportional incidence ratio; CI = confidence interval; cold season = winter and autumn; warm season = spring and summer.

Variable	*Mycoplasma bovis*				*Mycoplasma dispar*			
	Proportion of Positives to *M. bovis*	PI	PIR (95% CI)	*p*-Value	Proportion of Positives to *M. dispar*	PI	PIR (95% CI)	*p*-Value
Sampling season								
Cold season	14/33	42.4	1.1 (0.6–2.2)	0.76	9/33	27.3	1.4 (0.5–4.1)	0.50
Warm season	8/21	38.1			4/21	19.0		
Sex								
Male	14/30	46.7	1.4 (0.7–2.8)	0.34	9/30	30.0	1.8 (0.6–5.2)	0.28
Female	8/24	33.3			4/24	16.7		
Age								
≤5 months	11/28	39.3	0.9 (0.5–1.8)	0.82	11/28	39.3	5.1 (1.2–21.2)	0.02
>5 months	11/26	42.3			2/26	7.7		
Weigh								
≤175 Kg	12/27	44.4	1.2 (0.6–2.3)	0.58	9/27	33.3	0.4 (0.2–1.3)	0.13
>175 Kg	10/27	37.0			4/27	14.8		

## Data Availability

The original contributions presented in the study are included in the article/Appendix A, further inquiries can be directed to the corresponding authors.

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
