# Peer review of "Identification of *Mycoplasma* Species in Cattle Associated with Bovine Respiratory Disease Mortality"

_microorganisms, 2024, doi:10.3390/microorganisms12112340_

Round 1

Reviewer 1 Report

Comments and Suggestions for Authors

In the interesting manuscript Emanuele Carella et al. investigated the characteristics of Mycoplasma spp infections in cattle with Bovine Respiratory Disease (BRD) and the influence of various other factors was considered. And for the first time, Mycoplasma hyopharyngis isolated from the pharyngeal and nasal cavities of pigs was detected in the lungs of cattle with BRD. However, the quality of articles and tables needs to be strongly improved. With editing and some revisions, I feel that this manuscript will be suitable for publication.

1. Line 17-19: Please confirm the age of the positive cattle cases in years or months. This contradicts the contents of Table 1.

2. The Materials and Methods section is somewhat lengthy, and it is recommended that the statements be refined and simplified.

3. Why are there 2 pulmonary exudates and the others lung tissue in the samples collected in the 2.1 Sample collection section?

4. Line 128: In "50 mM of MgCl2", 2 should be a subscript, please correct.

5. Line 161: "MgSO4" has the same problem as "MgCl2".

6. The article lacks basic information on the sequences referenced from Genbank. Please supplement.

7. Figure 1: What conclusions can be drawn from the phylogenetic tree? Please supplement in the article.

8. Line 313-315: The article states that there were 10 cases of co-infection with E. coli, whereas in Table 1 there were 11 cases. Please confirm the number of cases again.

9. Line 343-345: The article states that there were a total of seven reported cases during the summer, but only six were introduced. Please provide a complete explanation.

10. Line 347-350: These two sentences are contradictory, please correct them.

11. Line 350-353: Why are ages categorised as over and under five years in the article but over and under five months in Table 1?

12. The discussion section is too wordy and unfocused, and Table 1 is difficult to reconcile with the content. Please simplify the statements and modify Table 1 to make it look more visual.

13. Table 2: Font formatting is inconsistent, please revise.

14. Line 580-582: Why is five months used as the age boundary in all the previous articles, but eight months is used as the boundary here?

15. Line 583-585: Same problem with Line 17-19.

16. In all the data statistics, “p” need to be capital and italic, please check the entire article for consistent formatting.

17. A total of forty-five references were cited for this article. Twenty-three references were from before 2015, of which five were from before 2000. References should be nearly 3-5 years. Please update references.

Author Response

Comments 1: Line 17-19: Please confirm the age of the positive cattle cases in years or months. This contradicts the contents of Table 1.

Response 1: The sentence has been corrected in the manuscript according to your comments (lines 17-19).

Comments 2: The Materials and Methods section is somewhat lengthy, and it is recommended that the statements be refined and simplified

Response 2: The Materials and Methods section has been revised in the manuscript.

Comments 3: Why are there 2 pulmonary exudates and the others lung tissue in the samples collected in the 2.1 Sample collection section?

Response 3: Pulmonary exudate was detected during the necropsy of specimen 39920/23, leading to a decision to sample it concurrently with the lung.

Comments 4: Line 128: In "50 mM of MgCl2", 2 should be a subscript, please correct

Response 4: The sentence has been revised in the manuscript according to your comment.

Comments 5: Line 161: "MgSO4" has the same problem as "MgCl2"

Response 5: The sentence has been revised in the manuscript according to your comment.

Comments 6: The article lacks basic information on the sequences referenced from Genbank. Please supplement.

Response 6: As requested, the sequences of the study have been deposited in GenBank. A table has also been created in the supplementary materials (Table S2) with the information regarding all the sequences used for the construction of the phylogenetic tree.

Comments 7: Figure 1: What conclusions can be drawn from the phylogenetic tree? Please supplement in the article.

Response 7: Since the genus Mycoplasma includes a large number of species, many of which are phylogenetically very similar, the construction of a phylogenetic tree can be a further means for the species identification of unknown isolates. Furthermore, the tree has allowed to quickly visualize any differences within the sequences belonging to the same species. The answers have been reported in the article as requested.

Comments 8: Line 313-315: The article states that there were 10 cases of co-infection with E. coli, whereas in Table 1 there were 11 cases. Please confirm the number of cases again.

Response 8: The number of co-infections with E. coli are 11 and the sentence has been revised in the manuscript (line 372).

Comments 9: Line 343-345: The article states that there were a total of seven reported cases during the summer, but only six were introduced. Please provide a complete explanation.

Response 9: The cases reported during the summer are 7 and the sentence has been revised in the manuscript (line 402).

Comments 10: Line 347-350: These two sentences are contradictory, please correct them.

Response 10: The paragraph was affected by a typo. The first sentence was modified in “At the univariable analysis for M. bovis, no variables presented a p-value <0.20, and thus, none were taken forward to the multivariable analysis.”. The rest of the paragraph has been restructured to improve readability (lines 411-415).

Comments 11: Line 350-353: Why are ages categorised as over and under five years in the article but over and under five months in Table 1?

Response 11: A mistake was made prior to submission to the journal. It is five months, not eight

Comments 12: The discussion section is too wordy and unfocused, and Table 1 is difficult to reconcile with the content. Please simplify the statements and modify Table 1 to make it look more visual

Response 12: Certainly, the discussion is rather extensive, but in order to provide a comprehensive commentary on our analyses and compare them to those of other articles, it is not possible to shorten this section of the article. If you believe there are any redundant parts or overlapping information, please point them out. Nevertheless, the discussion has been revised to provide detailed comments on all possible associations and sharpen its focus on the analyses performed.

Due to inconsistencies with the content, Table 1 has been replaced.

Comments 13:. Table 2: Font formatting is inconsistent, please revise.

Response 13: Table 2 has been formatted as indicated and two columns have been added to increase clarity (‘proportion of positives to M. bovis’ and ‘proportion of positives to M. dispar’). These two columns provide the figures used to calculate the proportional incidence.

Comments 14: Line 580-582: Why is five months used as the age boundary in all the previous articles, but eight months is used as the boundary here?

Response 14: An error was made before submission to the journal. It is five months, not eight.

Comments 15: Line 583-585: Same problem with Line 17-19.

Response 15: The sentence has been revised in the manuscript according to your comments.

Comments 16: In all the data statistics, “p” need to be capital and italic, please check the entire article for consistent formatting.

Response 16: The article was formatted as indicated: all ‘p’ values were replaced with ‘P’.

Comments 17: A total of forty-five references were cited for this article. Twenty-three references were from before 2015, of which five were from before 2000. References should be nearly 3-5 years. Please update references.

Response 17: Undoubtedly, references should be as recent as possible. Nevertheless, most articles with similar topics to ours, that have been published recently, have been cited. Since BRD is a disease with a long-standing impact, the most significant studies were published more than 5 years ago, but they are essential for data comparison and have therefore been included.

Reviewer 2 Report

Comments and Suggestions for Authors

The manuscript (microorganisms-3256863) aimed to characterize  Mycoplasma spp infections in cattle with Bovine Respiratory Disease (BRD), considering factors such as animal demographics, concurrent infections with other pathogens, post-mortem clinical findings and histological examinations, and seasonality. The manuscript has several promising aspects, but it requires revisions as several points need to be carefully revised before the next resubmission as follows:

1)      Abstract; Please add more information about the results obtained. Also, add a short conclusion at the end of the abstract.

2)      Clarify the hypothesis of the study at the end of the Introduction section.

3)      Line 112; specify the particular breed of cattle used in the study.

4)      Line 113-114; how was the sample size for the experiment determined and whether it was deemed appropriate for the study?

5)      Line 136 and 233; Include the full details for all instruments or kits used (i.e., company, city, country).

6)      Line 142-144; add a reference for the selection of these specific primers.

7)      Line 250; provide more details about the details of light microscopy used.

8)      How do you suggest the results of this study could impact current practices in cattle health management?

Author Response

Comments 1: Abstract; Please add more information about the results obtained. Also, add a short conclusion at the end of the abstract.

Response 1: The Abstract has been revised according to your comment.

Comments 2: Clarify the hypothesis of the study at the end of the Introduction section.

Response 2: The hypothesis of the study at the end of the Introduction has been clarified.

Comments 3: Line 112; specify the particular breed of cattle used in the study.

Response 3: The breed of the cattle analyzed was not a factor considered in this study. Therefore, the specific breeds of the cattle examined were not characterized.

Comments 4: Line 113-114; how was the sample size for the experiment determined and whether it was deemed appropriate for the study?

Response 4: A convenience sampling method was employed, wherein samples were obtained from cattle that that died from BRD and were subsequently submitted for necropsy to the Institute during the study period. Consequently, the sample size was not pre-specified but rather determined by the number of BRD cases presented to the Institute for necropsy during the study period.

Comments 5: Line 136 and 233; Include the full details for all instruments or kits used (i.e., company, city, country).

Response 5: The full details for all instruments or kits used have been included in the manuscript.

Comments 6: Line 142-144; add a reference for the selection of these specific primers

Response 6: The references have been added in the manuscript (Line 205, ref. 21).

Comments 7: Line 250; provide more details about the details of light microscopy used.

Response 7: The light microscopy used is ZEISS Axio Scope A1 and the detail has been added in the manuscript.

Comments 8: How do you suggest the results of this study could impact current practices in cattle health management?

Response 8: The study underscores the necessity of a comprehensive approach to cattle health management, incorporating age-specific strategies, advanced diagnostics and pathogen control. The implementation of these findings may result in more effective prevention and treatment of bovine respiratory disease, that may lead to an improved cattle health and productivity. Furthermore, Mycoplasma hyopharyngis has only been isolated from the nasal and pharyngeal mucosae of clinically normal swine. This study, for the first time, reports the detection of M. hyopharyngis in the pneumonic lung of a bovine with BRD in association with Proteus vulgaris. This finding suggests the potential for M. hyopharyngis to infect cattle lungs, although further research is necessary to elucidate its pathogenic potential. Therefore, this finding could impact current practices in cattle health management, as there are farms where both cattle and swine are raised. Transmission of Mycoplasma hyopharyngis could occur between these two livestock, leading to further economic loss

Reviewer 3 Report

Comments and Suggestions for Authors

The research submitted by Carella et al. aimed to identify the Mycoplasma species involved in cases of BRD, as well as the concomitant etiological agents. This research is important and contributes to a better understanding of the etiology of BRD in cattle within the sampled area and may be of interest to the entire veterinary medical community.

The abstract needs to be revised to reduce its introduction and include more significant results to attract readers.

The introduction is well-constructed and facilitates understanding of the study. However, several pieces of information are presented repetitively, and the authors should refine this section to make it more concise and direct, avoiding excessive repetition. Furthermore, the authors should include a sentence highlighting the novelty of the research.

The methodology section requires extensive revision for clarity, as well as the inclusion of numerous bibliographic references. Almost no references are used in this section. Additionally, the inclusion of a figure containing a flowchart of the experimental design will improve the quality and understanding of the manuscript.

The results section is poorly presented and tedious to read. The authors present a lot of information in isolation, likely because key results were included as supplementary material. These results should be incorporated into the main manuscript, which would reduce the excessive repetition of results. These findings should then be presented in the incorporated table (see Major remarks).

The discussion is an exhaustive presentation of data from other studies and comparisons between the numerical values obtained in this study. I suggest that the authors use the histopathological analyses more effectively to relate them to the primary etiological agents, as well as in cases of co-infection.

Major Remarks

·        Avoid excessive repetition of information in the introduction.

·        Revise the abstract to present more results of interest to the readers.

·        In section 2.1, the authors need to describe in detail how the sample collection was carried out.

·        In section 2.1, the authors mention that the samples were obtained from 46 farms. The authors should analyze whether there was a pattern of occurrence of etiological agents on the same properties. Did the etiological agents predominate within the same herd? This needs to be clarified in the manuscript.

·        The authors should reorganize the subsections of the Materials and Methods, creating a section titled “Identification of etiological agents,” followed by subsections for each agent. The current formatting is not adequate

·        In section 2.2, the authors could group the analyses that led to the identification of Mycoplasma species and emphasize that sequencing was performed to identify the species.

·        Additionally, the authors must include bibliographic references for all PCR analyses and for the entire Materials and Methods section to ensure reproducibility.

·        The authors should include information about the use of positive and negative controls for all molecular analyses.

·        Section 2.5 should be subdivided by agent following the manuscript’s presentation pattern and grouped with the methodologies presented in sections 2.6 and 2.7.

·        In section 2.9, the authors should describe the percentage similarity threshold used for species identification. Additionally, the authors should include a supplementary file with the origin of all microorganisms used in the phylogenetic tree, along with their respective GenBank identification numbers.

·        Table S2 should be revised and incorporated into the manuscript rather than remaining a supplementary file. The information presented in the table is essential for understanding, as it provides results on the etiological agents involved in cases of co-infection, as well as characterizing the pathological changes associated with each agent.

·        Lines 334-346: The authors should standardize the data by season and conduct appropriate statistical analyses to allow comparison of the syndrome's occurrence or its various etiological agents by season. The current data presentation is speculative, and the practical inferences are severely limited.

·        The table titles should be improved to be more comprehensive and self-explanatory.

·        Additionally, the tables need reformatting as they currently resemble frames, which is not appropriate for scientific presentation.

·        Figure 1 should be presented in high resolution, as its current quality is very poor.

·        At the end of the discussion, the authors should elaborate on the practical applications of their results for developing prophylactic measures for the syndrome studied.

·        The conclusion needs to be revised to objectively and precisely address the aim, which also needs reformulation. It should also include information on the occurrence of co-infections and risk factors.

·        The authors must provide access to the sequences obtained in GenBank. Have they been deposited? What are the accession numbers?

Minor Remarks

·        Use Mycoplasma with an initial capital letter and italicized in all mentions in the manuscript (including title, abstract, and all other sections).

·        The title needs complete revision as it is inconsistent with the presented study since the authors also detected various other viral etiological agents. Moreover, there was no characterization of Mycoplasma isolates, but rather species identification.

·        In the keywords, replace words already present in the title with others that will improve manuscript indexing.

·        Line 12: Replace “characterize” with “identify.”

·        Lines 16, 112, and elsewhere: Replace “succumbed.”

·        Lines 16/17: Include the total number and percentage of isolates identified, as well as the total evaluated.

·        Line 18: Replace the approximate fivefold risk value with the actual observed odds ratio, including its 95% confidence interval.

·        Line 23: Replace “suffering” with “infected.”

·        Lines 39, 79, and elsewhere: Replace “flora” with “microbiota.”

·        Lines 42, 89, and elsewhere: spp. should not be italicized.

·        Line 45: Specify which species of hosts.

·        Line 59: Replace “met” with a more suitable term.

·        Line 77: Ureaplasma is not a Mycoplasma.

·        Lines 105-108: The authors need to revise the objectives to align with the study, as they identified various viral agents, characterized lesions, and identified the species involved.

·        Line 111: Remove “opportunistically.”

·        Line 111: Replace “from” with “of.”

·        Lines 118-119: Specify how these data were obtained.

·        Line 214: Italicize all scientific names.

·        Line 220: Cite the MALDI-TOF reference.

·        Line 209: Which portion? How was it diluted? The methodology is unclear.

·        Section 2.8: Reference all methodologies in this section as well.

·        Line 252: No incidence values were presented. The authors should standardize the use of the term “frequency” as it is entirely different from the epidemiological term “incidence.”

·        Section 2.11: Include the software, manufacturer, and version.

·        Line 277: Include all percentage values in the manuscript, not just in this instance, but with all numerical values.

·        Line 394: Specie (singular).

Comments on the Quality of English Language

Minor/Moderate editing.

Author Response

Comments 1: The abstract needs to be revised to reduce its introduction and include more significant results to attract readers.

Response 1: The abstract has been revised according to your comment

Comments 2: The introduction is well-constructed and facilitates understanding of the study. However, several pieces of information are presented repetitively, and the authors should refine this section to make it more concise and direct, avoiding excessive repetition. Furthermore, the authors should include a sentence highlighting the novelty of the research.

Response 2: I don't understand what several pieces of information are presented repetitively in the introduction. could you be more specific? A sentence highlighting the novelty of the research has been added at the end of the introduction

Comments 3: The methodology section requires extensive revision for clarity, as well as the inclusion of numerous bibliographic references. Almost no references are used in this section. Additionally, the inclusion of a figure containing a flowchart of the experimental design will improve the quality and understanding of the manuscript.

Response 3: The methodology has been revised in the manuscript and the lacking bibliographic references has been added. The flowchart of the experimental design has been included

Comments 4: The results section is poorly presented and tedious to read. The authors present a lot of information in isolation, likely because key results were included as supplementary material. These results should be incorporated into the main manuscript, which would reduce the excessive repetition of results. These findings should then be presented in the incorporated table (see Major remarks)

Response 4: Table S2 has been incorporated into the manuscript, replacing table 1

Comments 5: The discussion is an exhaustive presentation of data from other studies and comparisons between the numerical values obtained in this study. I suggest that the authors use the histopathological analyses more effectively to relate them to the primary etiological agents, as well as in cases of co-infection.

Response 5: We appreciate your suggestion and have revised the discussion accordingly. However, we believe that a precise correlation between the histopathological lesions and the etiological agents may not be essential for the scope of this article, as the observed histological lesions are consistent with those typically associated with bovine respiratory diseases

Comments 6: In section 2.1, the authors need to describe in detail how the sample collection was carried out.

Response 6: The details of the sample collection has been added to the manuscript

Comments 7:  In section 2.1, the authors mention that the samples were obtained from 46 farms. The authors should analyze whether there was a pattern of occurrence of etiological agents on the same properties. Did the etiological agents predominate within the same herd? This needs to be clarified in the manuscript.

Response 7: Our findings indicate that no discernible pattern of etiological agent occurrence was observed within or across the 46 farms studied. This suggests that the distribution of etiological agents in these herds was largely random, with no clear evidence of clustering or predominance of specific agents within any given herd. This has been clarified in the manuscript

Comments 8: The authors should reorganize the subsections of the Materials and Methods, creating a section titled “Identification of etiological agents,” followed by subsections for each agent. The current formatting is not adequate

Response 8: The section materials and methods has been reorganized in the manuscript, according to your comment

Comments 9: In section 2.2, the authors could group the analyses that led to the identification of Mycoplasma species and emphasize that sequencing was performed to identify the species.

Response 9: End-point PCR was employed to detect Mycoplasma spp., after which the gel bands corresponding to the Mycoplasma spp. amplicons were excised and used for sequencing, allowing us to identify the Mycoplasma species. A sentence has been added to subsection 2.2.2. in response to your comment. The subsection 'Identification of Mycoplasma species and phylogenetic analysis' has been relocated to follow the subsection 'DNA extraction and PCR conditions for Mycoplasma spp.' in order to group related analyses focused on the identification of Mycoplasma species.

Comments 10: Additionally, the authors must include bibliographic references for all PCR analyses and for the entire Materials and Methods section to ensure reproducibility.

Response 10: The bibliographic references for all PCR analyses have been added to the Materials and Methods section.

Comments 11: The authors should include information about the use of positive and negative controls for all molecular analyses.

Response 11: The information about the use of positive and negative controls for all molecular analyses has been included in the manuscript.

Comments 12:  Section 2.5 should be subdivided by agent following the manuscript’s presentation pattern and grouped with the methodologies presented in sections 2.6 and 2.7.

Response 12: The section 2.5 has been revised in the manuscript, according to your comment.

Comments 13: In section 2.9, the authors should describe the percentage similarity threshold used for species identification. Additionally, the authors should include a supplementary file with the origin of all microorganisms used in the phylogenetic tree, along with their respective GenBank identification numbers.

Response 13: A similarity threshold of 97.0% was used for species identification; this information has been added in section 2.9, as requested.

We appreciate your suggestion to include a supplementary file with the origin of all microorganisms used in the phylogenetic tree, along with their respective GenBank identification numbers. However, the origin of most microorganisms detected in our study is already presented in the updated Table 1, where they are shown to primarily originate from the lungs of cattle with bovine respiratory disease (BRD). To further enhance the transparency of our data, we have added a supplementary table containing the GenBank identification numbers of all sequences used in the phylogenetic tree, without repeating the origin information that is already provided in Table 1.

Comments 14: Table S2 should be revised and incorporated into the manuscript rather than remaining a supplementary file. The information presented in the table is essential for understanding, as it provides results on the etiological agents involved in cases of co-infection, as well as characterizing the pathological changes associated with each agent.

Response 14: Table S2 has been incorporated into the manuscript, replacing table 1.

Comments 15: Lines 334-346: The authors should standardize the data by season and conduct appropriate statistical analyses to allow comparison of the syndrome's occurrence or its various etiological agents by season. The current data presentation is speculative, and the practical inferences are severely limited.

Response 15: As you rightly pointed out, seasonality can be an important risk factor for Mycoplasma infection. However, the analysis of the available data does not indicate the existence of any difference in the distribution of positives for M. bovis or M. dispar across the warm and cold season (Table 2). Unfortunately, the small sample size and data sparsity issues precluded any deeper investigation of the role of seasonality. In response to your comment, the existence of any difference in distribution of positives for M. bovis and M. dispar across seasonality was further investigated using the Fischer’s Exact. However, no significant differences were found (P=0.60). Given the apparent absence of any association between seasonality and positivity to M. bovis or M. dispar, and given the severe limitations due to the small sample size and data sparsity, the Authors deemed careful not to expand the analysis to avoid the risk of producing unreliable results.

Comments 16: The table titles should be improved to be more comprehensive and self-explanatory.

Response 16: The table’s titles have been revised. Furthermore, in Table 2 two columns have been added to increase clarity (‘proportion of positives to M. bovis’ and ‘proportion of positives to M. dispar’): these columns provide the figures used to calculate the proportional incidence.

Comments 17: Additionally, the tables need reformatting as they currently resemble frames, which is not appropriate for scientific presentation.

Response 17: Tables were formatted as required, following the journals’ article template.

Comments 18: Figure 1 should be presented in high resolution, as its current quality is very poor.

Response 18: The resolution of the figure 2 has been improved in the manuscript

Comments 19:  At the end of the discussion, the authors should elaborate on the practical applications of their results for developing prophylactic measures for the syndrome studied.

Response 19: A paragraph discussing the practical applications of our results has been added at the end of the discussion section.

Comments 20: The conclusion needs to be revised to objectively and precisely address the aim, which also needs reformulation. It should also include information on the occurrence of co-infections and risk factors.

Response 20: The conclusion has been revised in the manuscript according to your comment

Comments 21: The authors must provide access to the sequences obtained in GenBank. Have they been deposited? What are the accession numbers?

Response21: As requested, the sequences of the study have been deposited in GenBank. A table has also been created in the supplementary materials (Table S2) with the information regarding all the sequences used for the construction of the phylogenetic tree.

Comments 22: Use Mycoplasma with an initial capital letter and italicized in all mentions in the manuscript (including title, abstract, and all other sections).

Response 22: The term “Mycoplasma” has been rewritten according to your comment.

Comments 23: The title needs complete revision as it is inconsistent with the presented study since the authors also detected various other viral etiological agents. Moreover, there was no characterization of Mycoplasma isolates, but rather species identification.

Response 23: The title has been revised in the manuscript.

Comments 24:  In the keywords, replace words already present in the title with others that will improve manuscript indexing.

Response 24: The keywords present in the title have been replaced.

Comments 25: Line 12: Replace “characterize” with “identify.”

Response 25: The word “characterize” has been replaced in the manuscript (line 11).

Comments 26:  Lines 16, 112, and elsewhere: Replace “succumbed.”

Response 26: The word “succumbed” has been replaced throughout the manuscript.

Comments 27: Lines 16/17: Include the total number and percentage of isolates identified, as well as the total evaluated.

Response 27: The total number and percentage of isolates identified, as well as the total evaluated have been included in the abstract.

Comments 28: Line 18: Replace the approximate fivefold risk value with the actual observed odds ratio, including its 95% confidence interval.

Response 28: The proportional incidence ratio was added to the abstract and the sentence was slightly restructured to increase readability (lines 16-19).

Comments 28: Line 23: Replace “suffering” with “infected.”

Response 29: The word has been replaced in the manuscript (line 24).

Comments 29: Lines 39, 79, and elsewhere: Replace “flora” with “microbiota.”

Response 30: The word has been replaced in the manuscript.

Comments 30: Lines 42, 89, and elsewhere: spp. should not be italicized.

Response 31: The term has been modified according to your comment.

Comments 31: Line 45: Specify which species of hosts.

Response 32: The sentence has been modified according to your comment (line 48).

Comments 32: Line 59: Replace “met” with a more suitable term.

Response 33: The term “met” has been replaced in the manuscript (line 61-62).

Comments 33: Line 77: Ureaplasma is not a Mycoplasma.

Response 33: The sentence has been revised in the manuscript according to your comment (lines 79-82).

Comments 34: Lines 105-108: The authors need to revise the objectives to align with the study, as they identified various viral agents, characterized lesions, and identified the species involved.

Response 34: The sentence has been revised in the manuscript according to your comment (lines 107-115).

Comments 35: Line 111: Remove “opportunistically.”

Response 35: The term has been removed from the manuscript

Comments 36: Lines 118-119: Specify how these data were obtained.

Response 36: Upon arrival at our necropsy room, cattle carcasses were accompanied by an official veterinary report detailing the animal's demographic characteristics

Comments 37: Line 214: Italicize all scientific names.

Response 37: Staphylococcus aureus has been italicized in the manuscript

Comments 38: Line 220: Cite the MALDI-TOF reference.

Response 38: The reference has been added in the manuscript (line 157).

Comments 39: Line 209: Which portion? How was it diluted? The methodology is unclear.

Response 39: A portion of lung or trachea exhibiting macroscopic lesions was initially cauterized and then streaked onto Columbia Blood Agar and Gassner Agar. The samples were not diluted. The methodology has been explained in more details in the manuscript.

Comments 40: Section 2.8: Reference all methodologies in this section as well.

Response 40: The reference for the MALDI-TOF analysis has been added in this subsection. The other methodologies have not been referenced as they have been routinely used according to our internal standard operating procedures for over 30 years

Comments 41: Line 252: No incidence values were presented. The authors should standardize the use of the term “frequency” as it is entirely different from the epidemiological term “incidence.”

Response 41: Proportional incidence for Mycoplasma bovis was calculated as the ratio between the number of positives to M. bovis during the study period and the total number of positives to all Mycoplasma species. The proportional incidence was calculated for each group of the variables considered in the analysis (Table 2: sampling season, sex, age, weigh). For each variable, proportional incidences across different groups were compared through the calculation of the proportional incidence ratio. The same was done for Mycoplasma dispar. As mentioned above, two columns (‘proportion of positives to M. bovis’ and ‘proportion of positives to M. dispar’) have been added to show the figures used to calculate the proportional incidence.

As you rightly observed, there is a distinction between the concepts of frequency and incidence. Consequently, any comparison between these two concepts was carefully avoided in the manuscript

Comments 42: Section 2.11: Include the software, manufacturer, and version.

Response 42: This information has been included in Section 2.4. Statistical analysis (line 308).

Comments 43:  Line 277: Include all percentage values in the manuscript, not just in this instance, but with all numerical values.

Response 43: Percentage values have been included as required

Comments 44: Line 394: Specie (singular).

Response 44: The sentence has been revised

Reviewer 4 Report

Comments and Suggestions for Authors

Dear authors,

the manuscript „Characterization of mycoplasma Species in Cattle associated with Bovine Respiratory Disease mortality” is properly written. It widely characterizes mycoplasma species and its impact on animals. The manuscript will be adequate for publishing in Microorganisms if authors follow the reviewers’ comments. But there are some issues to discuss:

It is surprising that in this article only one species of mycoplasma was detected in each animal. The literature shows that these are mostly co-infections (J Vet Res 60, 391-397, 2016; DOI: 10.1515/jvetres-2016-00058). Please describe whether the detection of only 1 species of mycoplasma from 1 sample is due to the limitations of the PCR method. Please describe in more detail the principle of the PCR method used. How was the result obtained and analyzed, was the PCR product sequenced?

The M. bovis nomenclature has changed (https://www.vetbact.org/?artid=34): Phylum: Mycoplasmatota, Class: Mollicutes, Order: Mycoplasmoidales, Family: Metamycoplasmataceae; Genus: Mycoplasmopsis; Alternative Species Name(s):Mycoplasma bovis. Please consider at least to add in the bracket: “Alternative species name: Mycoplasmopsis bovis”.

Place the subchapter “microbiological investigations” after “Sample collection”.

Place the subchapter “histopatological investigation” after “microbiological investigations”.

Minor comments:

Authors should write “Mycoplasma ssp.” instead of “Mycoplasma ssp”.

Line 2: “mycoplasma” should be capitalized.

Line 51: Specify which antibiotic therapy.

Line 77: U. diversum is not a mycoplasma. Rebuild the sentence.

Line 188 and title of Table S1: Demographic data refers to people. Change into “population”.

Line 136: Add information about place of origin in the bracket.

Line 214: Write the name of bacteria with italic.

Line 274: Don’t write mycoplasma with italic in this place.

Figure 1: Not visible which data are reference and which derived from this study. Please mark it in the legend and in the dendrogram.

Author Response

Comments 1: It is surprising that in this article only one species of mycoplasma was detected in each animal. The literature shows that these are mostly co-infections (J Vet Res 60, 391-397, 2016; DOI: 10.1515/jvetres-2016-00058). Please describe whether the detection of only 1 species of mycoplasma from 1 sample is due to the limitations of the PCR method. Please describe in more detail the principle of the PCR method used. How was the result obtained and analyzed, was the PCR product sequenced?

Response 1: The finding of a single Mycoplasma species in all infected individuals is due to the limitations of the technique used: it is a genus-specific PCR, whose positive products were subsequently sequenced according to the Sanger method. It is likely that there were co-infections, which however were excluded from the study as they could not be sequenced directly from the DNA extracted using the PCR primers. The limitations of the technique, and consequently of the obtained results, have been highlighted in the manuscript.

Comments 2: The M. bovis nomenclature has changed (https://www.vetbact.org/?artid=34): Phylum: Mycoplasmatota, Class: Mollicutes, Order: Mycoplasmoidales, Family: Metamycoplasmataceae; Genus: Mycoplasmopsis; Alternative Species Name(s):Mycoplasma bovis. Please consider at least to add in the bracket: “Alternative species name: Mycoplasmopsis bovis”.

Response 2: Thank you for the update, but I do not understand where I should place the alternative species name. Can you specify it?

Comments 3: Place the subchapter “microbiological investigations” after “Sample collection”.

Response 3: The subchapter “microbiological investigations” has been relocated after “Sample collection”.

Comments 4: Place the subchapter “histopatological investigation” after “microbiological investigations”.

Response 4: The subchapter ‘Histopathological Investigation’ has been relocated before ‘Statistical Analysis’ due to the creation of a new section titled ‘Identification of Etiological Agents,’ which is followed by subsections detailing microbiological and molecular investigations for every etiological agent investigated

Comments 5: Authors should write “Mycoplasma ssp.” instead of “Mycoplasma ssp”."

Response 5: The term “Mycoplasma ssp.” has been revised in the manuscript.

Comments 6: Line 2: “mycoplasma” should be capitalized.

Response 6: The term “mycoplasma” has been revised in the manuscript.

Comments 7: Line 51: Specify which antibiotic therapy.

Response 7: The cited articles do not specify the antibiotic therapy.

Comments 8: Line 77: U. diversum is not a mycoplasma. Rebuild the sentence.

Response 8: The sentence has been revised according to your comment (lines 79-82).

Comments 9: Line 188 and title of Table S1: Demographic data refers to people. Change into “population”.

Response 9: The term “Demographic data” has been revised in the manuscript

Comments 10: Line 136: Add information about place of origin in the bracket.

Response 10: The information has been added in the manuscript

Comments 11: Line 214: Write the name of bacteria with italic.

Response 11: The term has been revised in the manuscript (line 142).

Comments 12: Line 274: Don’t write mycoplasma with italic in this place

Response 12: The term “mycoplasma” has been revised in the manuscript

Comments 13: Figure 1: Not visible which data are reference and which derived from this study. Please mark it in the legend and in the dendrogram.

Response 13: The figure (now Figure 2) has been amended to make it clear the distinction between data from the study and reference data.

Reviewer 5 Report

Comments and Suggestions for Authors

Summarize

The authors performed necropsies on cows that had collapsed due to respiratory disease on 46 farms in northwest Italia, and sampled 320 lung samples, 4 trachea, and 2 pulmonary exudates to analyze for mycoplasma spp infection. Mycoplasma genes were detected in samples from 54 of the 322 cows, and were classified into 9 types based on their homology with database sequences. The most common species detected were M. bovis and M.disper, with the others being few in number. In particular, M. bovigenitalium and M. hyopharyngis were detected in only one individual each. The authors conclude that this is the first time M. hyopharyngis has been detected in the lung of cattle. The authors also analyzed the association between mycoplasma-positive samples and co-infecting microorganisms, season, sex, age, and weight for each species, and claimed that M. disper infection was approximately five times more likely in cattle under 5 years/months of age than older than 5 years/months of age.

Comments

This was a large-scale sampling survey, and I respect the efforts of the authors. However, the reported results and conclusions confirm previous data from other countries and others, and are therefore lacking in novelty. In addition, the conclusions are flawed due to a lack of data, and it is thought that the influence of sampled individuals and the environment is greater than the differences between microorganisms, so the content is not sufficient to interest readers who are interested in microorganisms themselves. In terms of reader interest, I would recommend submitting the paper as a report to a dairy or veterinary journal.

The following points particularly require improvement.

1.  In Segment 35 and 58: 5 years old is probably a 5 month old mistake.

2.  There are no information about breed of cattle. Different breed respond significantly differently to microorganisms, so it is critical to demonstrate that these are not biased in your samples.

3.  You claimed that M. disper infection was approximately five times more likely in cattle under 5 years/months of age than older than 5 years/months of age. I feel that you need to discuss the fact that the risk of infection is decreasing among the older, rather than the higher risk of infection among the younger.

4.  The data information is fragmented and cannot be verified. It is desirable to make all data, including infection information for each individual, verifiable as a single table or data set.

5.  Although it is claimed that this is the first detection of M. hyopharyngis in bovine lungs, it is based on a single sample and only PCR sequence analysis. If this is to be taken as a conclusion, isolation and culture should be performed to confirm the infection.

Author Response

Comments 1: This was a large-scale sampling survey, and I respect the efforts of the authors. However, the reported results and conclusions confirm previous data from other countries and others and are therefore lacking in novelty. In addition, the conclusions are flawed due to a lack of data, and it is thought that the influence of sampled individuals and the environment is greater than the differences between microorganisms, so the content is not sufficient to interest readers who are interested in microorganisms themselves. In terms of reader interest, I would recommend submitting the paper as a report to a dairy or veterinary journal.

Response 1: Our results confirm those reported in other studies, but with the difference that we observed the presence of M. hyopharyngis in the lung of a bovine dead from BRD. In addition, this study highlights the notable prevalence of Mycoplasma bovis, which stands out as a primary pathogen associated with bovine respiratory disease (BRD). In contrast, other species identified in our research are often suggested to act as secondary invaders or opportunistic agents. This distinction is crucial because it underscores the varying roles these microorganisms play in disease dynamics. However, the environment plays a significant role in the transmission and spread of microorganisms. Factors such as housing conditions, hygiene practices, and exposure to other animals can impact the prevalence of Mycoplasma species within a herd.

The conclusion was implemented in the manuscript.

Comments 2:  In Segment 35 and 58: 5 years old is probably a 5 month old mistake.

Response 2: A careless mistake was made which was corrected in the manuscript

Comments 3: There are no information about breed of cattle. Different breeds respond significantly differently to microorganisms, so it is critical to demonstrate that these are not biased in your samples.

Response 3: The breed of the cattle analyzed was not a factor considered in this study. Therefore, the specific breeds of the cattle examined were not characterized.

Comments 4: You claimed that M. dispar infection was approximately five times more likely in cattle under 5 years/months of age than older than 5 years/months of age. I feel that you need to discuss the fact that the risk of infection is decreasing among the older, rather than the higher risk of infection among the younger.

Response 4: This finding has been commented in the manuscript, according to your comment.

Comments 5: The data information is fragmented and cannot be verified. It is desirable to make all data, including infection information for each individual, verifiable as a single table or data set.

Response 5: Table S2 has been inserted into the manuscript replacing Table 1

Comments 6: Although it is claimed that this is the first detection of M. hyopharyngis in bovine lungs, it is based on a single sample and only PCR sequence analysis. If this is to be taken as a conclusion, isolation and culture should be performed to confirm the infection

Response 6: Indeed, the first detection of M. hyopharyngis in a single case was observed via PCR coupled with sequencing. This approach does not allow us to determine whether this Mycoplasma species was viable, but given that the bovine died from BRD, its involvement cannot be ruled out. Nevertheless, I have revised the conclusion emphasizing that isolation and cultivation would be useful to confirm the role of M. hyopharyngis in BRD.

Round 2

Reviewer 1 Report

Comments and Suggestions for Authors

Thank you!

Reviewer 3 Report

Comments and Suggestions for Authors

My suggestions were fully adressed.

Reviewer 5 Report

Comments and Suggestions for Authors

Comment

 I understand that the authors were trying to answer my comments, but I still have questions about important points. And also, I am concerned that the number of samples and information available are too small to draw significant conclusions that will satisfy readers of Microorganisms.

 The answer that there is no information on cattle breeding because it is not used as a factor in the analysis is really disappointing because it cannot be proven that the sample is not biased. I want to make sure that at least all of the cattle are beef cattle.

 In line 638, I do not understand how it is important from an epidemic or academic perspective to know which Mycoplasma species are most prevalent in the Piedmont region. I also question whether it is appropriate to investigate only the proportion of BRD deaths caused by these species, which have different infection and transmission risks.

 Table S1 should include the data from new Table 1 so that the relationship between infection species and age can be seen at a glance. It is preferable to supplement it with a table like Excel data. I also found that one author's old attached comment remains in table S1, and I agree with that opinion, given that there is no difference between municipalities or provinces at present.